# Towards Quantifying Long-Range Interactions in Graph Machine Learning: A Large Graph Dataset and a Measurement

**Huidong Liang**[*]    **Haitz Sáez de Ocáriz Borde**[*]    **Baskaran Sripathmanathan**[*]

**Michael Bronstein**    **Xiaowen Dong**

University of Oxford

## Abstract

Long-range dependencies are critical for effective graph representation learning, yet most existing datasets focus on small graphs tailored to inductive tasks, offering limited insight into long-range interactions. Current evaluations primarily compare models employing global attention (e.g., graph transformers) with those using local neighborhood aggregation (e.g., message-passing neural networks) without a direct measurement of long-range dependency. In this work, we introduce `City-Networks`, a novel large-scale transductive learning dataset derived from real-world city road networks. This dataset features graphs with over $10^5$ nodes and significantly larger diameters than those in existing benchmarks, naturally embodying long-range information. We annotate the graphs based on local node eccentricities, ensuring that the classification task inherently requires information from distant nodes. Furthermore, we propose a generic measurement based on the Jacobians of neighbors from distant hops, offering a principled quantification of long-range dependencies. Finally, we provide theoretical justifications for both our dataset design and the proposed measurement—particularly by focusing on over-smoothing and influence score dilution, which establishes a robust foundation for further exploration of long-range interactions in graph neural networks. Our *dataset* and *measurement* are available on *PyTorch Geometric*, and the code for reproducing the experimental results can be found at *github.com/LeonResearch/City-Networks*.

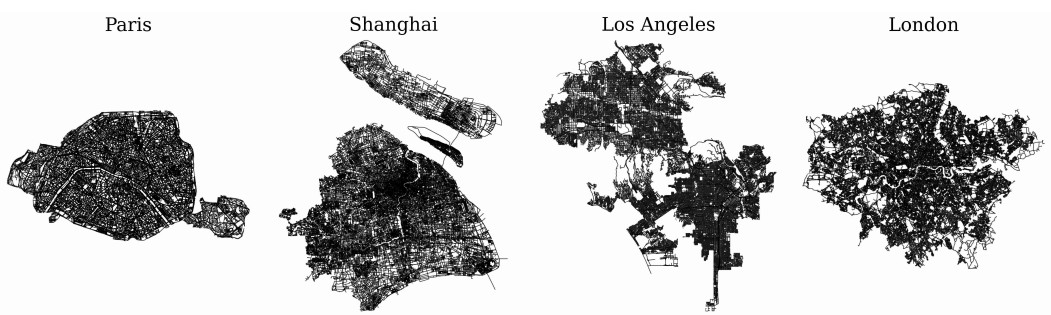

Figure 1: Visualizations of `City-Networks` for Paris, Shanghai, Los Angeles, and London.

## 1 Introduction

Graphs are a widely used mathematical abstraction across nearly every branch of science. They are particularly effective for modeling the intricate and non-uniform interactions found in real-world data, where nodes stand in for objects and edges depict their connections. The growing recognition of the versatility of graph representations has sparked intense interest in Graph Neural

---

[*]Equal contribution, contact `huidong.liang@eng.ox.ac.uk`

Networks (GNNs) (Scarselli et al., 2009; Wu et al., 2021), driving innovation in deep learning for both geometric and graph-centric applications. Most GNNs, in particular variants based on a message-passing mechanism (Gilmer et al., 2017), exchange information between one-hop neighbors per layer to build node representations. While they found wide success in analysing citation and social networks (Yang et al., 2016b), one significant limitation concerns their capability of handling long-range dependencies, where interactions between distant nodes might be required to solve a task. Most existing datasets are not sufficient in assessing this: for instance, social networks, despite comprising thousands of nodes, often exhibit the *small world* effect (Watts & Strogatz, 1998) with short average path lengths and high clustering coefficients. The node labels on these graphs usually possess a homophilic pattern where nodes tend to connect with "similar" or "alike" others (McPherson et al., 2001), making it possible to propagate sufficient information for modelling with only a few message-passing layers. On the other hand, while connected nodes tend to have different properties on heterophilic graphs (Zhu et al., 2020; Ma et al., 2022), solving the tasks in those cases do not necessarily require the handling of long-range dependencies (Arnaiz-Rodriguez & Errica, 2025).

Recently, Long Range Graph Benchmark (`LRGB`) (Dwivedi et al., 2023) introduces alternative graph datasets based on super-pixels and molecules with larger diameters than those of the previous works. To justify the existence of long-rangeness, the authors compare classical GNNs with Graph Transformers (GTs), which leverage global attentions over the entire graph, and associate the observed performance gaps with the presence of long-range dependencies. However, conclusions that are mainly derived from empirical comparisons may not be reliable as they can be largely influenced by hyperparameter tuning (Tönshoff et al., 2024), leading to an ambiguous assessment of the long-range interactions. Moreover, `LRGB` and other synthetic long-range benchmarks (Bodnar et al., 2021; Zhou et al., 2025) all focus on inductive learning tasks with relatively small graphs—typically on the order of 10 to $10^2$ nodes, while currently there is no long-range benchmark that considers large graphs with real-world topology for transductive learning. This is a critical gap in the literature since applying GTs (Rampášek et al., 2022), which are expected to better capture long-range interactions, to large graphs is significantly more challenging compared to small-graph inductive tasks due to the computational complexity in its positional encoding and global attention (Borde, 2024).

We aim to address these limitations in this work, and our main contributions are as follows:

- We propose `City-Networks`, a transductive learning dataset that consists of four large-scale city road networks with a topology distinct from those commonly found in the literature. In particular, it features grid-like large graphs with up to $500k$ nodes and diameters of up to $400$, where the labels are annotated based on node accessibility that inherently requires long-range dependency in its calculation. To the best of our knowledge, this is the first large-graph dataset designed for testing long-range dependencies in graph representation learning.

- We empirically test classical GNNs and GTs on our dataset under different model depths, and compare their behaviors to those on other common graph datasets that are short-ranged, large-scale, heterophilic, or long-range dependent. The results on our datasets, unlike those on the existing datasets, suggest that communication with neighbors from distant hops consistently improves the performance of all models, supporting the presence of long-range signals.

- To quantify such long-range dependency, we further introduce a generic[1] measurement that quantifies *per-hop influence* of a focal node's neighbors on its prediction, which is estimated by the aggregated $\ell_1$-norm of the Jacobian from a trained model for the task at hand at each hop around the focal node. This per-hop analysis of the task range goes beyond the concurrent work of Bamberger et al. (2025) and provides novel insights: we observe that distant hops exert a greater influence on all baseline models in our `City-Networks` compared to those on other commonly used graph datasets in the literature.

- Finally, we theoretically justify the graph structure in our dataset from a spectral perspective on over-smoothing, whose rate we link to the algebraic connectivity and diameter of the graph. In addition, we relate our proposed measurement to the concept of influence as defined in the literature, and study the dilution of mean influence score in grid-like graphs to justify our design.

---

[1] Here, generic means that the metric can be applied to any differentiable GNN or Graph Transformer, without requiring architectural modifications or assumptions about the model class.

Table 1: Statistics of `City-Networks` compared to common graph datasets, where $d$, $C$, $T$, $diam$, $homo$ represent degree, clustering coefficient, transitivity, diameter, and homophily, respectively.

| Dataset | #Nodes | #Edges | avg($d$) | std($d$) | max($d$) | $C$ | $T$ | $diam$ | $homo$ |
|---|---|---|---|---|---|---|---|---|---|
| Paris | $114k$ | $183k$ | 3.2 | 0.8 | 15 | 0.03 | 0.03 | 121 | 0.70 |
| Shanghai | $184k$ | $263k$ | 2.9 | 1.0 | 8 | 0.04 | 0.04 | 123 | 0.75 |
| Los Angeles | $241k$ | $343k$ | 2.8 | 1.0 | 9 | 0.04 | 0.05 | 171 | 0.75 |
| London | $569k$ | $759k$ | 2.7 | 1.0 | 10 | 0.04 | 0.05 | 404 | 0.76 |
| Cora | $2.7k$ | $5.3k$ | 3.9 | 5.2 | 168 | 0.24 | 0.09 | 19 | 0.81 |
| ogbn-arxiv | $169k$ | $1.16m$ | 13.7 | 68.6 | $13k$ | 0.23 | 0.02 | 25 | 0.65 |
| Amazon-Ratings | $24k$ | $93k$ | 7.6 | 6.0 | 132 | 0.58 | 0.31 | 46 | 0.38 |
| PascalVOC-SP | 479 | $1.3k$ | 5.7 | 1.2 | 10 | 0.43 | 0.40 | 28 | 0.92 |

## 2 THE CITY-NETWORKS DATASET

In this section, we begin by identifying the limitations and challenges in current literature. Then, we characterize how the topology of `City-Networks` differs from the existing datasets, and proceed to justify the rationale behind our long-range labeling strategy. Lastly, we explain how our dataset addresses the current limitation and discuss the new challenges brought to the field.

**Challenges in testing long-range dependency.** To design and fairly evaluate graph datasets of long-range dependency, we need to address three research challenges:

1. How can we generate long-range signals in a principled and controllable manner, so that models are required to communicate with sufficiently distant neighbors of a node to predict its label?

2. Beyond the small graphs used in `LRGB` or other synthetic benchmarks, how can we design long-range signals on large real-world networks to test the scalability of a model?

3. How can we define a principled measure to quantify and compare the level of long-rangeness across different datasets?

We will discuss how our proposed dataset handles the first two challenges in this section, and address the third challenge in Section 4.

### 2.1 LARGE-SCALE ROAD NETWORKS WITH LONG DIAMETERS

**Real-world network topologies and features.** As shown in Figure 1, our `City-Networks` consists of street maps in four cities: Paris, Shanghai, Los Angeles, and London, which are obtained by querying OpenStreetMap (Haklay & Weber, 2008) with OSMnx (Boeing, 2024). In particular, we consider a city network inclusive of all types of roads in the city (e.g., *drive*, *bike*, *walk*, etc.), where nodes represent road junctions with features like *longitude*, *latitude*, *land use*, etc.; and edges represent road segments with features like *road length*, *speed limit*, *road types*, etc. Next, to facilitate a typical node classification task, we apply a simple neighborhood aggregation that transforms edge features into node features by averaging the features of incidental edges and then concatenating them to the features of the focal node. These new node features represent, for instance, the average speed limit around a road junction or the probability of finding a residential road nearby. As a result, the final dataset contains 37 node features, where we also make the graphs undirected so that they only represent connectivity between junctions, and retain the largest connected component in each city. We refer readers to Appendix C.1 for a complete list of features and our approach in dataset processing.

**Large graphs with long diameters and low maximum degrees.** We can observe from Table 1 that our datasets feature large diameters from $100$ to $400$, which are much longer than those of the common graphs used in the literature. In particular, compared to the super-pixel graphs in `PascalVOC-SP` with typically 500 nodes, our datasets have much larger sizes ranging from $100k$ to $500k$. Meanwhile, compared to social networks such as `Cora`, `ogbn-arxiv`, and `Amazon-Ratings`, our `City-Networks` have smaller average clustering coefficients and much lower maximum degrees. We argue that this distinct graph topology enables us to effectively design learnable long-range signals on graphs, as explained in the next paragraph and further justified theoretically in Section 5.

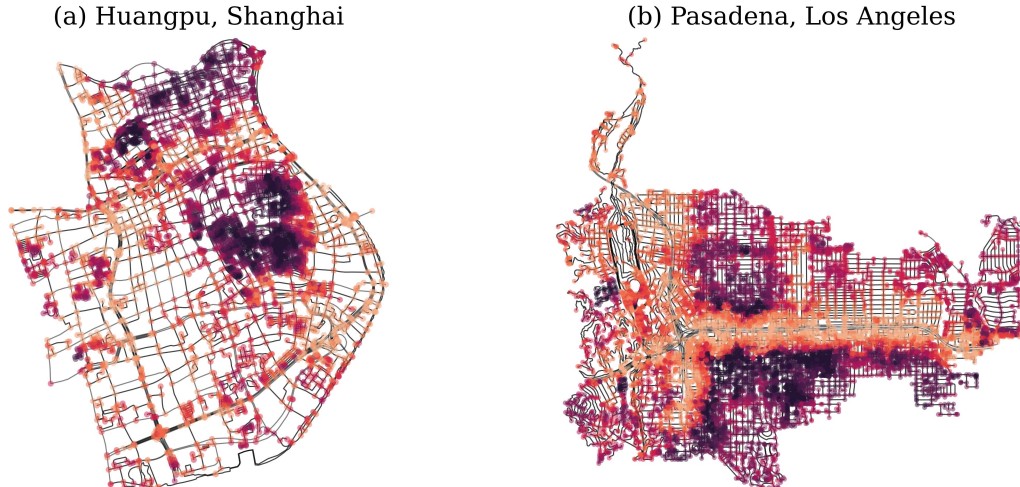

(a) Huangpu, Shanghai  (b) Pasadena, Los Angeles

Figure 2: Visualizations of node accessibility estimations based on local eccentricity in two sub-regions, where **darker** colors indicate **smaller** eccentricity values, i.e., nodes that are easier to access.

## 2.2 LONG-RANGE LABELS BASED ON URBAN ACCESSIBILITY

Based on the city road networks, we design a real-world task that requires the handling of long-range dependencies. Specifically, we aim to predict how accessible a road junction is based on its own location as well as neighboring characteristics of the urban landscape. This is a useful task especially when it comes to evaluating urban design principles such as the 15-minute city (Abbiasov et al., 2024). To quantify accessibility, we measure the distance one needs to travel from one road junction to its $k$-hop neighbors along road segments in the road network. By design, solving this task requires the model to be able to access information within the $k$-hop neighborhood of each focal node and, by adjusting $k$, we can design a long-range task as desired. Such a notion of accessibility is directly related to node eccentricity $\varepsilon(v)$ in network science (Newman, 2018), which looks at the maximum distance from $v$ to all other nodes in the graph $\mathcal{G} = (V, E)$.

However, using the exact eccentricity as signals is not ideal for two reasons. First, we cannot control the level of long-rangeness, as eccentricity is determined by the entire graph structure and not adjustable by design. In particular, although predicting $\varepsilon(v)$ appears to require distant information based on its definition, in many cases (Figure 7, Appendix C.2), the underlying signal can be well inferred by spatial features (e.g., geographical coordinates) alone. This is because nodes in the city center generally have lower eccentricity due to the grid-like topology in road networks. Second, the calculation of eccentricity requires computing all pair-wise shortest paths in the network, which is typically at cost $\mathcal{O}(|V|^3)$ and is hence prohibitively expensive for labeling our large-scale networks.

**Node classification task based on local eccentricity.** With the above consideration in mind, we propose a local eccentricity measure $\hat{\varepsilon}_k(v)$ that only considers $k$-hop neighbors $\mathcal{N}_k(v)$ of node $v$ when calculating the longest shortest paths with the following expressions:

$$\hat{\varepsilon}_k(v) = \max_{u \in \mathcal{N}_k(v)} \rho_w(v, u), \quad \rho_w(v, u) = \min_{\pi \in P(v,u)} \sum_{e \in \pi} w(e), \tag{1}$$

where $\rho_w(v, u)$ is the weighted shortest path distance from node $v$ to node $u$, $P(v, u)$ is the set of all possible paths from node $v$ to node $u$, and $w(e)$ denotes the edge weight for edge $e$ (we use road length for this particular purpose, while in all other cases, the graphs are considered unweighted). To design a classification task, we then split $\hat{\varepsilon}_k(v)$ for all nodes into 10 quantiles which we use as node labels for the classification task.

We emphasize that this local approach is introduced not only for computational efficiency, but it also brings in two important considerations. First, *it creates long-range signals that extend to $k$-hop neighbors by design, i.e., a known "ground-truth" range*, hence is directly controllable and testable. Second, *it ensures both graph topology and node features contribute to the modelling*, avoiding the problem of high correlation between node signal with either the graph structure or node features

alone. Indeed, since $k$ and the edge weights used to compute $\hat{\varepsilon}_k(v)$ are unknown to the model, it must integrate structural information from distant neighborhoods with their spatial information (geographic location, road type, land use, etc.) to infer the long-range signal, which makes the task non-trivial. We provide a deeper analysis in Appendix C.3 regarding the roles of graph structure and spatial features.

**Choice of the long-range level $k$.** To assess long-range dependency, $k$ should be sufficiently large to distinguish our setting from short-range tasks (e.g., social graphs with typical message passing around 4 hops). Thanks to the large grid-like topology with long diameters, we can create long enough node signals based on different local networks. At the same time, $k$ should also not be too large such that (i) the selected neighborhoods maintain local characteristics; and (ii) baseline models under a $k$-layer architecture, which captures the required information, can fit into common GPUs for fair benchmarking purposes. After experimenting with values from 8 to 32 (Figure 9, Appendix C.4), we design the task at $k$=16, which strikes a balance between sufficient long-rangeness and computational efficiency. We refer readers to Appendix C.4 for a more detailed justification of our methods.

**Visualizations and interpretations.** As shown in equation 1, calculating $\hat{\varepsilon}_k(v)$ naturally requires information from distant neighbors up to $k$ hops, and it carries a practical meaning which relates to the accessibility of node $v$ in the network. As an illustration, we visualize two sub-regions from the city maps in Figure 2 with $k$=16. We can observe that nodes on major transportation routes such as freeways and highways tend to have larger eccentricities than those in populated areas. This is because reaching a node's 16-hop neighbors from a highway junction often requires traveling a much longer distance compared to road junctions in a downtown area. It is also clear from Figure 2 that a significant part of the graph topology is grid-like and possibly quasi-isometric to a lattice, as we will later discuss in Section 6. Note that the homophily scores, as reported in Table 1, are reasonably high since nodes with similar $\hat{\varepsilon}_k$ tend to cluster together.

**New challenges for learning long-range dependencies.** The proposed dataset presents a significant challenge for both GNNs and GTs. In general, such long-range dependencies create a fundamental trade-off for graph learning: while increasing the number of layers in GNNs helps propagate information from distant hops, it also leads to issues such as over-smoothing (Li et al., 2018; Nt & Maehara, 2019; Rusch et al., 2023), over-squashing (Alon & Yahav, 2021; Topping et al., 2022; Di Giovanni et al., 2023), and vanishing gradients (Arroyo et al., 2025). On the other hand, GTs, which rely on attention mechanisms with quadratic computational complexity, face scalability challenges when applied to our large-scale city networks compared to learning on smaller long-ranged graphs like those in `LRGB` (Dwivedi et al., 2023) and other synthetic benchmarks (Zhou et al., 2025).

# 3   BENCHMARKS: FROM CLASSICAL GNNS TO GRAPH TRANSFORMERS

In this section, we benchmark classical GNNs and GTs at different numbers of message-passing layers on `City-Networks`, and then contrast their behaviors with results on other commonly used graphs to examine the long-range dependencies across datasets.

**Experimental setups.** We consider transductive node classification with train/validation/test splits of $10\%/10\%/80\%$ on all `City-Networks`, in which we evaluate various classical GNNs and GTs. Specifically, we benchmark GCN (Kipf & Welling, 2017), GraphSAGE (Hamilton et al., 2017), GAT (Veličković et al., 2018), and GCNII (Chen et al., 2020) for GNNs; and GraphGPS (Rampášek et al., 2022), Exphormer (Shirzad et al., 2023), and SGFormer (Wu et al., 2023) for GTs. In addition, an MLP is also included to reflect the importance of graph topology (or lack thereof) in our task. Lastly, we fix the data split for benchmarking purposes and report the mean and standard deviation of the test accuracy over 5 runs with different random seeds.

**Evaluation protocols.** As a hypothesis, *when a graph model is predicting on a certain node $v$, it will benefit from communicating with $v$'s neighbors from distant hops if the task is long-ranged.* Following this rationale, we evaluate baseline models at different numbers of layers $L = [2, 4, 8, 16]$, with the expectation that a larger $L$ would lead to better performance after sufficient training. By comparison, we also evaluate the same baselines on four representative datasets from the literature: `Cora`, `ogbn-arxiv`, `Amazon-Ratings`, and `PascalVOC-SP` from `LRGB`, which are homophilous, large-scale, heterophilous, and long-range dependent, respectively. We then compare the behaviors of baseline models across different datasets while increasing the number of layers $L$.

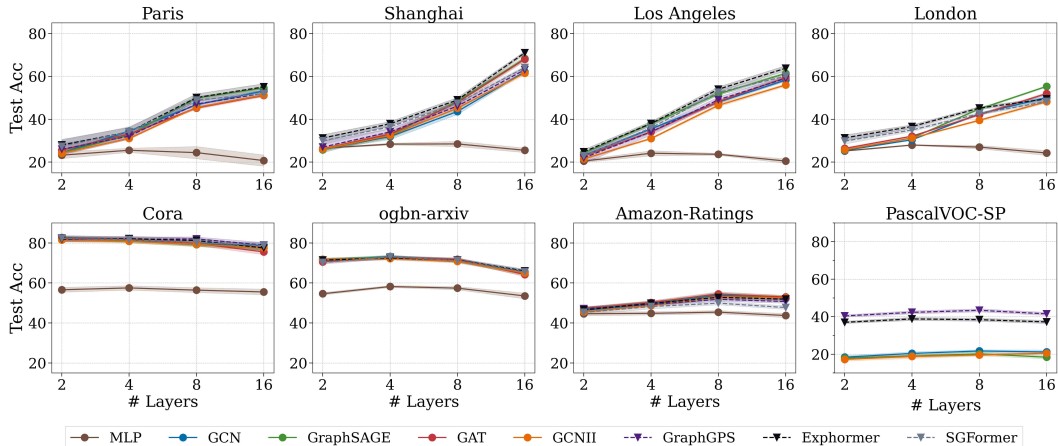

Figure 3: Baseline results across datasets at different number of layers $L = [2, 4, 8, 16]$. The results for GraphGPS are not shown on `London` as it is Out-of-Memory on our 48GB GPU; the result for SGFormer on `PascalVOC-SP` is also not reported as it's not originally designed for inductive setting.

For fair comparison purposes, we follow the latest GNN tuning technique from Luo et al. (2024) that considers residual connection, normalization, dropout, etc. Readers are also referred to Appendix D.2 for full details on our training and experimental setups.

**Results.** From Figure 3, we can observe a clear and consistent improvement in performance for all baselines on our `City-Networks` as their message-passing depth $L$ increases from 2 to 16. It suggests that, *even with sufficient training, a shallow graph model cannot outperform its deeper counterpart if it does not communicate with sufficiently distant hops*. This result also means that the gains from incorporating long-range information as $L$ increases outweigh other side effects of depth. By contrast, the performance of baseline models gradually degrades as $L$ increases on `Cora` and `ogbn-arxiv` due to the locality of these tasks and depth-related issues. To elaborate on this matter, we further provide a theoretical justification for our dataset in Section 5, where we link the rate of over-smoothing (a prominent issue associated with depth) to the algebraic connectivity and network diameter of the underlying graph. On the other hand, the baseline performance remains relatively unchanged with only a slight increase on `Amazon-Ratings` and `PascalVOC-SP`, which suggests that these tasks are relatively short-ranged compared to `City-Networks`. To examine this hypothesis, we therefore introduce a quantitative measure of long-range dependency in the next section and compare it across datasets and baselines.

Table 2: Best baseline results ($L = 16$) on `City-Networks`.

| Baseline | Type | Paris | Shanghai | Los Angeles | London |
|---|---|---|---|---|---|
| | MLP | $25.5 \pm 0.4$ | $28.4 \pm 0.6$ | $24.1 \pm 0.5$ | $27.9 \pm 0.1$ |
| | ChebNet | $54.1 \pm 0.2$ | $66.5 \pm 0.1$ | $61.4 \pm 0.4$ | $54.7 \pm 0.2$ |
| | GCN | $53.2 \pm 0.3$ | $62.1 \pm 0.2$ | $58.3 \pm 0.3$ | $50.1 \pm 0.7$ |
| MPNNs | GraphSAGE | $54.6 \pm 0.2$ | $68.3 \pm 0.5$ | $61.4 \pm 0.3$ | $55.4 \pm 0.2$ |
| | GAT | $51.1 \pm 0.3$ | $68.0 \pm 0.5$ | $59.5 \pm 0.3$ | $52.0 \pm 0.3$ |
| | GCNII | $51.3 \pm 0.2$ | $61.5 \pm 0.4$ | $56.0 \pm 0.3$ | $48.2 \pm 0.3$ |
| | DropEdge | $48.2 \pm 0.2$ | $60.8 \pm 0.4$ | $55.5 \pm 0.3$ | $45.0 \pm 0.3$ |
| | GraphGPS | $52.1 \pm 0.6$ | $63.0 \pm 0.5$ | $59.8 \pm 0.5$ | OOM |
| GTs | Exphormer | $55.1 \pm 0.8$ | $70.2 \pm 0.4$ | $63.8 \pm 0.6$ | $49.5 \pm 0.4$ |
| | SGFormer | $52.0 \pm 0.8$ | $64.1 \pm 0.3$ | $60.1 \pm 0.7$ | $48.3 \pm 0.3$ |

For completeness, we summarize the best baseline performance on `City-Networks` in Table 2, and further conducte an ablation study in Appendix D.3 that (i) tests baselines under different hidden_size; and (ii) fixes the model size at $L = 16$, refrains the model from seeing beyond hop-$H$ at each node, then tests baselines at different $H$. The results are all consistent with our above findings, and the potential limitation of our experiments is discussed in Appendix F.

## 4   A LONG-RANGE MEASUREMENT BASED ON JACOBIANS

In this section, we propose to quantify the long-range dependency of a task given a trained model, by evaluating how the *influence score* $I(v, u)$ between node-pairs (Xu et al., 2018; Gasteiger et al., 2022) varies with distance. The influence score measures the sensitivity of a GNN layer at node $v$ to the input feature of node $u$ using the Jacobian: $I(v, u) = \sum_i \sum_j \left| \partial \boldsymbol{H}_{vi}^{(L)} / \partial \boldsymbol{X}_{uj} \right|$, where $\boldsymbol{H}_{vi}^{(L)}$ is the $i^{th}$ entry of node $v$'s embedding at layer $L$, and $\boldsymbol{X}_{uj}$ is the $j^{th}$ entry of the input feature vector for node $u$. Unless otherwise specified, we assume $L$ refers to the last logit layer.

Based on the influence score, we define the *average total influence* $\bar{T}_h$ at $h^{th}$ hop as:

$$T_h(v) = I_{\text{sum}}(v, h) = \sum_{u:\rho(v,u)=h} I(v, u), \quad \bar{T}_h = \frac{1}{N} \sum_v I_{\text{sum}}(v, h), \tag{2}$$

where $T_h(v) = I_{\text{sum}}(v, h)$ is the *total influence* from the $h^{th}$-hop neighbors of node $v$, $\rho(v, u)$ is the length of the shortest path between $v$ and $u$ (note that this is equivalent to the $h$-hop shell later discussed in Section 6), and $N$ is the number of nodes in the network. Also, we would like to highlight that when $h = 0$, $T_0(v) = I(v, v)$ becomes the influence of the feature at node $v$ on its output. The average total influence quantifies how much, on average, the features of nodes that are $h$ hops away affect the output at the focal node. In other words, by summing the Jacobian-based influence scores from all $h$-hop neighbors and then averaging over all nodes, it provides an *expected measure of the cumulative effect that distant nodes have on each node as the focal node*.

Given equation 2, we further define the average size $R$ of the *influence-weighted receptive field* as:

$$R = \frac{1}{N} \sum_{v \in V} \frac{\sum_{h \geq 0}^{H} h \cdot T_h(v)}{\sum_{h \geq 0}^{H} T_h(v)}, \tag{3}$$

where $H$ is the maximum number of hops to be considered. Intuitively, one can understand $R$ as measuring *how far away the average unit of influence is*, since the influences from distant nodes to the target node will be proportionally stronger in long-range tasks compared to those in short-range tasks. Note that the proposed $R$ bears similarity to the recent work of Bamberger et al. (2025): the definition of $R$ in equation 3 is based on the shortest path distance, which corresponds to a specific instance of the measure in Bamberger et al. (2025). However, unlike that work which focuses on an aggregated measure of the range, our work has a particular focus on using the *per-hop influence* $T_h(v)$ as a diagnostic tool of the dependency decay, which leads to the key analysis in Figure 4 below.

Finally, we provide an analysis of the computation cost of our measurement in Appendix E and discuss its potential limitations on large dense graphs (a common challenge for Jacobian-based methods (Xu et al., 2018; Gasteiger et al., 2022) in the literature) in Appendix F.

**Results.** We validate $R$ and $\bar{T}_h$ on `City-Networks` using the baselines at $L = 16$ layers from Section 3. Since `PascalVOC-SP` is under an inductive setting, we randomly sample 100 graphs from its testing set (more than $400k$ nodes in total) and report their average. The results in Table 3 show that $R$ is consistently higher on our `City-Networks` across all models compared to those on the other graph datasets, indicating a longer effective range of influence.

Table 3: Average size of the influence-weighted receptive field $R$ across different datasets and models.

| Model | Paris | Shanghai | L.A. | London | Cora | ogbn-arxiv | Amazon | PascalVOC |
|---|---|---|---|---|---|---|---|---|
| GCN | 4.86 | 5.55 | 5.36 | 6.09 | 2.56 | 1.34 | 1.92 | 3.38 |
| GraphSAGE | 4.92 | 5.73 | 5.44 | 5.97 | 2.37 | 1.40 | 1.80 | 2.99 |
| GCNII | 3.62 | 3.64 | 3.68 | 3.66 | 2.76 | 3.5 | 1.79 | 2.60 |
| GraphGPS | 8.18 | 7.88 | 7.92 | OOM | 2.65 | OOM | 6.86 | 7.14 |
| Exphormer | 5.71 | 7.06 | 7.15 | 7.90 | 2.84 | 2.80 | 1.43 | 1.42 |
| SGFormer | 3.75 | 4.25 | 4.03 | 4.01 | 3.21 | 1.21 | 2.46 | NA |

To better compare $\bar{T}_h$ across models and datasets, we normalize it by $\bar{T}_0$ and report $\bar{T}_h/\bar{T}_0$ in Figure 4. For all models, we generally observe a rapid influence drop at distant hops on `Cora` and `ogbn-arxiv`, whereas the decay is noticeably much slower on `City-Networks`. The patterns on `Amazon-Ratings` and `PascalVOC-SP` generally fall between the above two cases with

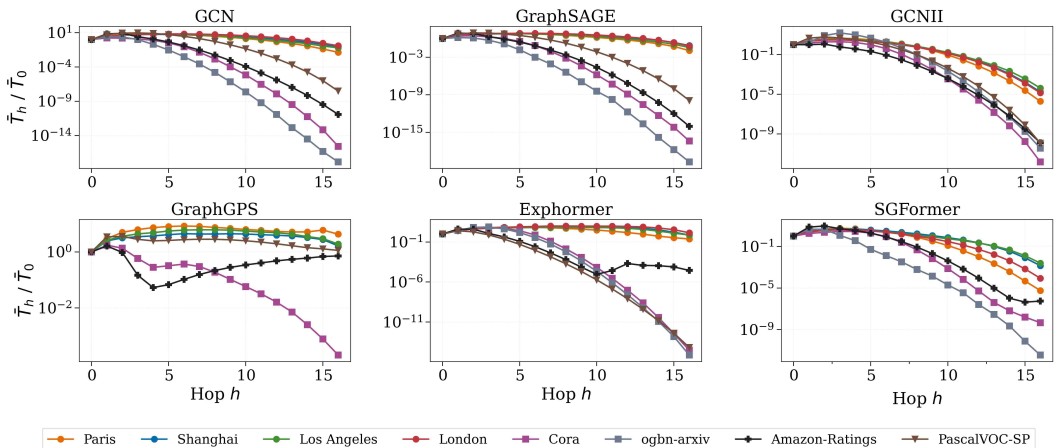

Figure 4: Normalized average total influence $\bar{T}_h/\bar{T}_0$ averaged across nodes at different distances. Note that the influence calculation for GraphGPS is Out-of-Memory on `London` and `ogbn-arxiv`.

influence concentrated on the first few hops, which corroborate our findings in Section 3. While $R$ and $\bar{T}_h$ are both model-dependent, the above results yield a consistent ordering across datasets: models trained on `City-Networks` are required to leverage information from more distant hops to perform well, compared to those trained on existing datasets.

Note that the bias in the models will naturally lead to a biased estimation of the ground-truth range using the proposed measurement, and we will discuss this limitation in more detail in Appendix F. In addition, we further provide results on `RingTransfer` (Bodnar et al., 2021), a synthetic experiment for testing long-range dependency using small ring-like graphs under inductive settings in Appendix D.4; and provide results on other common heterophilic datasets in Appendix D.5.

For the rest of the paper, we will first provide a theoretical justification of the proposed dataset by linking over-smoothing to algebraic connectivity and network diameter in Section 5, and then present the intuition behind the proposed long-range measurement in Section 6.

## 5    THEORETICAL JUSTIFICATION OF TOPOLOGIES IN CITY-NETWORKS

In this section, we provide a spectral analysis of over-smoothing in GNNs to justify our dataset design. As such, our main goal is not an in-depth theoretical analysis, since both over-smoothing and spectral analysis of GNNs have been carried out extensively in the literature (Rusch et al., 2023; Wu et al., 2023; Oono & Suzuki, 2020; Rong et al., 2019); instead, our aim is to build on top of these and derive simple results which provide a theoretical grounding of our dataset construction. The section is organized as follows. First, we review the concept of information loss in the limit of infinite GNN layers due to over-smoothing, which effectively vanishes the original node features and makes representations collapse to a value that is only dependent on the graph topology (Section 5.1 and Appendix G.1). Next, we relate the rate at which over-smoothing happens and the eigenvalues of the normalized adjacency operator in linearized GCNs (Section 5.2). Given that eigenvalues with larger absolute values slow down over-smoothing, we analyze how the magnitude of the eigenvalues relates to the graph topology of our datasets. This provides justification that graphs with large diameters promote bigger positive eigenvalues for the normalized adjacency operator and hence are less vulnerable to over-smoothing (Theorem 5.2 and Appendix G.3). Finally, we argue that less over-smoothing makes it possible for GNNs to capture long-range dependencies in the case of our proposed dataset. All proofs are presented in Appendix G.2.

### 5.1    PRELIMINARIES: A SPECTRAL PERSPECTIVE ON OVER-SMOOTHING FOR LINEAR GCNS

We begin by analyzing over-smoothing through the lens of linearization, following the framework established in Wu et al. (2019) to build Simple Graph Convolutional Networks (SGCs). We select this

Table 4: The lower bound for $\lambda_{N-1}(\tilde{\boldsymbol{S}}_{adj})$ on different datasets.

| Dataset | Paris | Shanghai | L.A. | London | Cora | arxiv | Ratings | Pascal |
|---|---|---|---|---|---|---|---|---|
| Lower Bound | 0.4741 | 0.6344 | 0.6095 | 0.5921 | 0.0324 | -0.0639 | 0.1224 | 0.4857 |

model because it is a linearized and more interpretable version of the widely adopted GCN model, which has also been used for theoretical analysis in previous works such as (Giovanni et al., 2022).

Let $\mathcal{G} = (V, E)$ be an undirected graph with $N = |V|$ nodes, adjacency matrix $\boldsymbol{A} \in \mathbb{R}^{N \times N}$ and degree matrix $\boldsymbol{D} \in \mathbb{R}^{N \times N}$ ($\boldsymbol{D}_{ii} = \sum_j \boldsymbol{A}_{ij}$). The normalized (augmented) adjacency operator $\tilde{\boldsymbol{S}}_{adj} \in \mathbb{R}^{N \times N}$ is defined as $\tilde{\boldsymbol{S}}_{adj} = (\boldsymbol{I} + \boldsymbol{D})^{-\frac{1}{2}}(\boldsymbol{I} + \boldsymbol{A})(\boldsymbol{I} + \boldsymbol{D})^{-\frac{1}{2}}$, where the original adjacency matrix and degree matrix have been augmented with self-loops. Wu et al. (2019) discuss the spectral properties of the normalized adjacency operator, which satisfies: $\lambda_N = 1$, and $|\lambda_i| < 1$ for all $i < N$. Therefore $\lim_{l \to \infty}(\tilde{\boldsymbol{S}}_{adj})^l = \tilde{\boldsymbol{u}}_N \tilde{\boldsymbol{u}}_N^T$, where $\tilde{\boldsymbol{u}}_N$ is the eigenvector corresponding to $\lambda_N$, whose entries at node $v$ are $\sqrt{1 + \text{degree}(v)}$.

Via linearization (detailed in Appendix G.1), as the number of layers tends to infinity, the graph convolution operation collapses all node features to scalar multiples of $\tilde{\boldsymbol{u}}_N$, resulting in complete loss of the original feature information, that is, the learned representations suffer from *over-smoothing*.

## 5.2 Over-smoothing Rate on Algebraic Connectivity, Diameter, and Sparsity

We now provide intuition about the types of graph topologies that would mitigate the over-smoothing problem. In particular, sparse graphs—those lacking the small-world effect commonly found in citation networks—and graphs with large diameters tend to experience over-smoothing at a slower rate. We argue that this slower rate of over-smoothing implies a higher likelihood that GNNs can learn long-range dependencies during optimization. We validate this intuition by bounding the eigenvalues of the normalized adjacency matrix using graph properties, then showing how these eigenvalues influence over-smoothing, and thus how controlling these graph properties can reduce the likelihood of over-smoothing. More concretely, we extend (Wu et al., 2019, Theorem 1), which only covers $\lambda_1$ and $\lambda_N$, to Proposition 5.1 which we prove in Appendix G.2, covering the normalized algebraic connectivity $\lambda_{N-1}$ and showing that $\lambda_{N-1}(\boldsymbol{S}_{adj}) < \lambda_{N-1}(\tilde{\boldsymbol{S}}_{adj})$.

**Proposition 5.1** (Self-loops decrease algebraic connectivity of the original graph). Assume a connected graph $\mathcal{G}$ with more than two nodes. For all $\gamma > 0$,

$$\lambda_{N-1}(\boldsymbol{S}_{adj}) = \lambda_{N-1}\left(\boldsymbol{D}^{-\frac{1}{2}}\boldsymbol{A}\boldsymbol{D}^{-\frac{1}{2}}\right) < \lambda_{N-1}\left((\gamma\boldsymbol{I} + \boldsymbol{D})^{-\frac{1}{2}}(\gamma\boldsymbol{I} + \boldsymbol{A})(\gamma\boldsymbol{I} + \boldsymbol{D})^{-\frac{1}{2}}\right). \quad (4)$$

This allows us to relate the second largest positive eigenvalue of $\tilde{\boldsymbol{S}}_{adj}$ to the topology of graphs in the proposed dataset, using bounds similar to those in (Chung, 1997, Lemma 1.14).

**Theorem 5.2** (Bound on second largest positive eigenvalue of the normalized adjacency operator). Let $d_{max}$ be the maximum degree of a vertex in $\mathcal{G}$, and $diam(\mathcal{G})$ the diameter of $\mathcal{G}$, which must be $diam(\mathcal{G}) \geq 4$. Then

$$\lambda_{N-1}(\tilde{\boldsymbol{S}}_{adj}) > \frac{2\sqrt{d_{max} - 1}}{d_{max}} - \frac{2}{diam(\mathcal{G})}\left(1 + \frac{2\sqrt{d_{max} - 1}}{d_{max}}\right). \quad (5)$$

From Table 4, we see that the lower bound in equation 5 is generally higher for our datasets; this is because the bound is decreasing in $d_{max}$ and increasing in $diam(\mathcal{G})$, showing graphs with large diameter and low maximum degree will be more resilient to over-smoothing. We further provide the rationale behind this lower bound in Appendix G.3, as well as its relations to the Braess paradox and previous study (Jamadandi et al., 2024) regarding over-smoothing and graph sparsity in Appendix G.4.

## 6 Theoretical Justification of Jacobian-Based Measurement

Finally, we justify the proposed measurement for analyzing long-range interactions in GNNs, particularly by comparing the mean and the sum of influence scores of neighboring nodes as candidates

to represent the *total influence score* for a focal node $v$, $T_h(v)$. Recall that $T_h(v) = I_{\text{sum}}(v, h) = \sum_{u:\rho(v,u)=h} I(v, u)$ as in equation 2. We first introduce definitions for the standard lattice in Euclidean space, $h$-hop shells, and quasi-isometric graphs, and then provide a mathematical analysis of *influence score dilution*. Due to space limits, we provide the definitions in Appendix I.

**Lemma 6.1** (Growth of $h$-hop shells in grid-like graphs). Let $\mathcal{G} = (V, E)$ be a graph that is *grid-like* in $D$ dimensions (e.g., let us presume the graphs in the proposed dataset are a subgraph of $\mathbb{Z}^D$ or quasi-isometric to it, which seems reasonable given Figure 2) and assume the node degrees are uniformly bounded. Then, there exist positive constants $\mathfrak{C}_1$ and $\mathfrak{C}_2$ (depending on $D$ and the local geometry of $\mathcal{G}$) and an integer $h_0$ such that for all $h \geq h_0$,

$$\mathfrak{C}_1 h^{D-1} \leq |\mathcal{N}_h(v)| \leq \mathfrak{C}_2 h^{D-1}. \tag{6}$$

After having quantified the growth of the $h$-hop shell, one can prove the following theorem, which motivates our aggregation choice.

**Theorem 6.2** (Dilution of mean aggregated influence in grid-like graphs). Suppose that for a fixed $v$ and for each $h \geq h_0$ there exists a distinguished node $u^* \in \mathcal{N}_h(v)$ with a strong influence on $v$, quantified by $I(v, u^*) = I^* > 0$, while for all other nodes $u \in \mathcal{N}_h(v) \setminus \{u^*\}$ the influence $I(v, u)$ is negligible (i.e. zero). Define $I_{\text{sum}}(v, h) = \sum_{u \in \mathcal{N}_h(v)} I(v, u) = \sum_{u:\rho(v,u)=h} I(v, u)$ and $I_{\text{mean}}(v, h) = \frac{1}{|\mathcal{N}_h(v)|} \sum_{u \in \mathcal{N}_h(v)} I(v, u)$. Then, $I_{\text{sum}}(v, h) \geq I^*$, and $I_{\text{mean}}(v, h) \leq \dfrac{I^*}{\mathfrak{C}_1 h^{D-1}}$. Hence, as $h \to \infty$, we have $I_{\text{mean}}(v, h) \to 0$, while $I_{\text{sum}}(v, h)$ remains bounded below by $I^*$. This also holds for planar graphs, where $D = 2$.

**Corollary 6.3** (Dilution for a planar grid-like graph). The dilution of the mean aggregated influence for a planar grid-like graph (like our city networks) is proportional to $\frac{1}{h}$.

**Corollary 6.4** (Faster dilution over aggregated $h$-hop neighborhoods). Let $\mathcal{G}$ be a grid-like graph in $D$ dimensions, and define the aggregated $h$-hop neighborhood (or ball) of a node $v$ as $B_h(v) = \bigcup_{i=1}^{h} \mathcal{N}_i(v)$. As before, suppose that within $B_h(v)$ there exists a unique node $u^*$ with influence $I(u^*, v) = I^* > 0$ and that for all other nodes $u \in B_h(v) \setminus \{u^*\}$, the influence is negligible (i.e. zero). Then, the mean aggregated influence is diluted at a rate proportional to $1/h^D$. In particular, for a planar graph ($D = 2$), the dilution occurs at a rate proportional to $1/h^2$.

The analysis above formally justifies the choice in equation 2, which considers the aggregate influence of neighboring nodes $T_h(v) = I_{\text{sum}}$ as a more reliable measure than the mean $I_{\text{mean}}$, which is susceptible to dilution, particularly in the case of distant neighbors. This is key to the computation of the *average total influence* in equation 2 and the *influence-weighted receptive field* in equation 3. Finally, note the following:

**Corollary 6.5** (The dilution problem does not affect the average total influence). Let $T_h(v) = I_{\text{sum}}(v, h)$ be the total influence from the $h$-hop neighborhood $\mathcal{N}_h(v)$ of node $v$, and let $\overline{T}_h = \frac{1}{|V|} \sum_{v \in V} T_h(v)$ be the average total influence over all nodes in $V$. Suppose that for every $v$ and every $h$ there exists at least one distinguished node $u^* \in \mathcal{N}_h(v)$ satisfying $I(v, u^*) \geq I^* > 0$. Then, $\overline{T}_h \geq I^*$, $\forall h$.

Proofs are provided in Appendix J, and a more detailed justification of Jacobian-based influence score and connections to existing literature, along with its limitations, can be found in Appendix H.

# 7 CONCLUSION

The main objective of our work is to provide better tools to help quantify long-range interactions in GNNs. Previous benchmarks, such as the LRGB (Dwivedi et al., 2023), are introduced in the context of small graph inductive learning, using solely the performance gap between classical GNNs and GTs to support the presence of long-range signals. In this work, we introduce a new large graph dataset based on city road networks, featuring long-range dependencies for transductive learning, and propose a principled measurement to quantify such dependencies. We also provide theoretical justification of both the proposed dataset and measurement, focusing on over-smoothing and influence score dilution. Beyond benchmarking purposes, our work also holds potential for a broader impact, informing applications in urban planning and transportation by providing tools to analyze and predict accessibility within city road networks.

ACKNOWLEDGMENT

Huidong Liang is funded by the ESRC Grand Union Doctoral Training Partnership and the Oxford-Man Institute of Quantitative Finance.

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

## A    DATASET AND CODE AVAILABILITY

Our dataset, **_CityNetwork_**, and our long-range measurement, **_total_influence_**, are both publicly available on *PyTorch Geometric* (PyG 2.7.0) under `torch_geometric.datasets` and `torch_geometric.utils`. We also open-sourced the code for dataset construction and benchmarking at *https://github.com/LeonResearch/City-Networks*.

## B    NETWORK STATISTICS

In this section, we explain the statistics used in Table 1 that characterize our dataset and discuss the estimation approach used when direct computation is impractical. We consider undirected graphs, $\mathcal{G} = (V, E)$ with $N = |V|$ nodes and $|E|$ edges, where each node $v$ is associated with a feature vector $\boldsymbol{X}_v \in R^D$ and a label $y_v \in \mathcal{Y}$ from a finite class set.

**Node degree.**    The degree $\deg(v)$ of a node $v$ represents the number of edges adjacent to it. To characterize the distribution of node degree in a network, we consider its mean $\mu_k$ and standard deviation $\sigma_k$:

$$\mu_k = \frac{1}{N} \sum_{v \in V} \deg(v), \quad \sigma_k = \sqrt{\frac{1}{N} \sum_{v \in V} (\deg(v) - \mu_k)^2}. \tag{7}$$

Importantly, networks of different topologies will have different degree distributions. For example, social networks often exhibit a power-law property, where a few nodes have high degrees while most nodes have relatively low degrees. As a result, their degree distribution tends to follow a scale-free pattern with a high degree variance $\sigma_k^2$. In contrast, grid-like networks, such as city networks and super-pixel graphs, have a structured layout of connections, leading to a more uniform degree distribution with lower variance.

**Clustering coefficient.**    The clustering coefficient $C_v$ measures the tendency of a node $v \in V$ to form a tightly connected group based on triangles, and its average $\bar{C}$ measures the overall level of clustering:

$$C_v = \frac{2T_v}{\deg(v)(\deg(v) - 1)}, \quad \bar{C} = \frac{1}{N} \sum_{v \in V} C_v, \tag{8}$$

where $T_v$ is the number of triangles that include node $v \in V$. Alternatively, *transitivity* offers a global measure of clustering with the following expressions:

$$Transitivity = \frac{3 \times \text{number of triangles}}{\text{number of connected triples}}. \tag{9}$$

Intuitively, social networks tend to have a high *average clustering coefficient* and *transitivity* due to their community structure with highly connected hubs and frequent triadic closures. On the other hand, our city networks exhibit a low clustering coefficient and transitivity, as their structured and sparse connectivity (e.g., forming lattices) reduces the prevalence of triangles. Note that `LRGB`, although being "grid-like", shows an even higher $\bar{C}$ and transitivity than social networks. This is because its networks contain many triangles, as it uses semantic super-pixels as nodes with pixel borders being the edges.

**Diameter**    The diameter $D$ of a network is the longest shortest path between any two nodes:

$$D = \max_{u,v \in V} \rho(u, v), \tag{10}$$

where $\rho(u, v)$ is the shortest path length from node $u$ to node $v$. Note that in this case the distance is unweighted for a more direct comparison with other datasets. It represents the maximum communication delay in the network and varies significantly across different network structures. In social networks, the presence of hubs greatly reduces the average shortest path length, leading to a small-world effect with a relatively small diameter. In contrast, grid-like networks lack hubs, and their regular structure causes the diameter to increase more rapidly as the network size grows.

Since the exact calculation of $D$ has an impractical computational complexity at $\mathcal{O}(N^2 \log(N))$, we use the following approach to estimate the approximate diameter $\hat{D}$ of our city networks. For

all nodes in a given city, we select the ones with the maximum and minimum latitude and longitude, respectively: $coord(v_1) = ( \cdot , \text{lat}_{max})$, $coord(v_2) = ( \cdot , \text{lat}_{min})$, $coord(v_3) = (\text{long}_{max}, \cdot )$, and $coord(v_4) = (\text{long}_{min}, \cdot )$. Based on these, we compute the shortest path between $(v_1, v_2)$ and $(v_3, v_4)$, and take their maximum as our final diameter estimate, that is, $\hat{D} = \max\big(d(v_1, v_2), d(v_3, v_4)\big)$. Note that the exact diameter $D$ will always be larger than our estimation $\hat{D}$.

**Homophily.** The node homophily score (Pei et al., 2020) quantifies the tendency of nodes with the same label to be connected:

$$Homo = \frac{1}{N} \sum_{v \in V} \frac{|\{u \in \mathcal{N}(v) \,|\, y_u = y_v\}|}{|\mathcal{N}(v)|}, \tag{11}$$

where $\mathcal{N}(v)$ is the set of neighbors of node $v$, and $y_v$ represents the label of node $v$. A higher homophily indicates a stronger preference for connections between nodes of the same class, which can significantly impact the performance of GNN models.

## C  DATASET DETAILS

### C.1  RAW FEATURE PROCESSING

This section provides additional details of the features and labels of our `City-Networks`.

**Node and edge features.** The node and edge features in our dataset are derived from real-world features provided by OpenStreetMap for both road junctions and road segments, as detailed below.

- Three numerical features for the road junctions (nodes):
    - *latitude*: the latitude of the current road junction.
    - *longitude*: the longitude of the current road junction.
    - *street count*: the number of connected roads in both directions.
- One categorical features for the road junctions (nodes):
    - *land use*: the type of land use at the current coordinate: *residential*, *industrial*, *forest*, *farmland*, *commercial*, *railway*, etc.
- Two numerical features for road segments (edges):
    - *road length*: the length of the road in meters.
    - *speed limit*: the speed limit on the current road in km/h.
- Two binary features for road segments (edges):
    - *one-way*: if the current road can only be used in one direction by vehicles.
    - *reversed*: if the current road alternates between different directions during rush hours in the morning and evening, which is also sometimes called "tidal flow".
- Two categorical features for road segments (edges):
    - *lanes*: number of lanes in the current road, which takes either numerical values (e.g. $1, 2, 3, ...$), or a list of numerical values (e.g. $[1, 2], [2, 3], [4, 5], ...$) when the current road has different number of lanes at different segments. We treat this feature as a categorical variable during modeling.
    - *road type*: the type of the current road, with possible values being: *service*, *residential*, *footway*, *primary*, *secondary*, *tertiary*, etc.

**Feature engineering.** Since the categorical features *land use*, *lanes*, and *road type* contain many categories of only a few data points and varies across different networks, we only take the top $8$ categories with most entries for each categorical feature, and treat the rest as a single category - *other*. Based on our observations, we can cover more than 90% of the network with the top $8$ categories in all cases. This strategy leads to 12 node features and 25 edge features after one-hot encoding. Next, to facilitate a typical node classification task, we apply a simple neighborhood aggregation that

transforms edge features into node features by averaging the features of incidental edges and then concatenating them to the features of the focal node. These new node features represent, for instance, the average speed limit around a road junction or the probability of finding a residential road nearby. As a result, the final dataset contains 37 node features after processing. Lastly, we transform the graph into an undirected one by merging all edges between each pair of nodes into a single edge, where the edge features are averaged using `to_undirected(reduce="mean")` from PyG.

## C.2 Long-range Node Labels based on Local Eccentricity

**Controllable long-range signal based on local eccentricity.** As mentioned in Section 2.2, we use a $k$-hop ego-network to estimate the eccentricity of each node. Specifically, after obtaining the neighborhood $\mathcal{N}_k(v)$ within $k$ hops of node $v$, we compute the shortest path from node $v$ to all nodes within this neighborhood, and take the maximum as the local eccentricity $\hat{\varepsilon}_k(v)$:

$$\hat{\varepsilon}_k(v) = \max_{u \in \mathcal{N}_k(v)} \rho_w(v, u), \quad \rho_w(v, u) = \min_{\pi \in P(v, u)} \sum_{e \in \pi} w(e), \tag{12}$$

where $\rho_w(v, u)$ is the weighted shortest path distance from node $v$ to node $u$, $P(v, u)$ denotes the set of all possible paths from node $v$ to node $u$, and $w(e)$ represents the edge weight for edge $e$. Here, we use one of the edge features *road length* as the edge weight $w(e)$, such that the local eccentricity $\hat{\varepsilon}_k(v)$ will indicate the maximal traveling distance (in meters) from node $v$ to its $k$-hop neighbors. Figure 5 shows the distributions of such approximation at $k = 16$ across different city networks, in which `Paris` and `Shanghai` have the most skewed and uniform distributions, respectively.

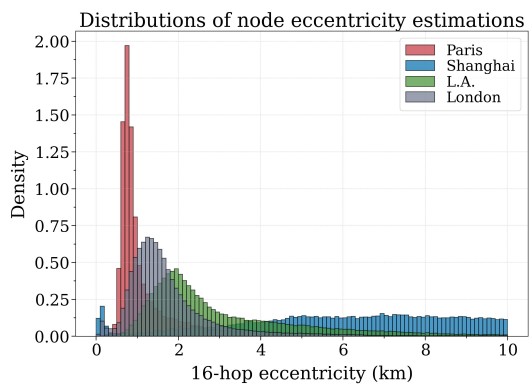

Figure 5: Distribution of the 16-hop eccentricity for all nodes in each of our `City-Networks`.

Importantly, this method allows us to know a priori that the long-range signal should be highly correlated with the hop $k$, which facilitates us in task design and model benchmarking. Lastly, after obtaining the local eccentricity for all nodes, we split them into 10 quantiles as the final node labels for transductive classification.

**Discussions on more realistic estimations of accessibility.** While our approach gives us a clean and controllable setup to study long-range topological interactions, it inevitably neglects some of the complexities inherent in real-world transportation dynamics, in which other factors like road speed, capacity, and traffic congestion strongly influence how "accessible" one area is from another. However, we believe our framework can be naturally extended by using road length and speed limit to approximate travel time when defining the edge weight in the eccentricity from equation 1. Here we present some preliminary results on Paris under this strategy. The results in Figure 6 suggest a similar rising trend in performance as we increase the model's

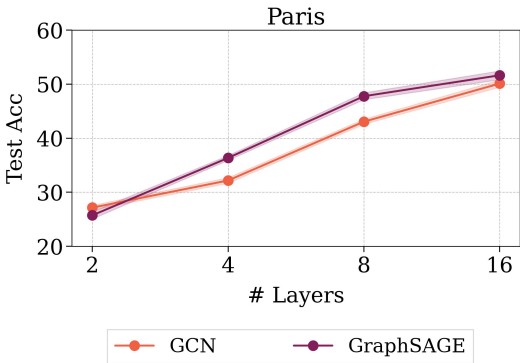

Figure 6: Results on `Paris` where labels are derived with travel time being the edge weight.

depth, which is consistent with our statements in the main text. We note that other travel-time proxies, such as data from the Google Maps API, may provide a closer approximation to real-world conditions, and we leave such directions for future work.

**Computational cost.** The exact eccentricity typically requires $\mathcal{O}(|V|^3)$ using the Floyd-Warshall algorithm (Floyd, 1962). However, with our local approach, the computation is at a much cheaper cost

of $\mathcal{O}(|\hat{E}| + |\hat{V}| \log |\hat{V}|)$ using the Dijkstra's algorithm (Dijkstra, 1959), where $|\hat{V}|$ and $|\hat{E}|$ are the average number of nodes and edges across all $k$-hop ego-networks. The calculation is implemented with `networkx` (Hagberg et al., 2008) on a CPU cluster of 72 Intel Xeon @ 2.3GHz cores, which takes 3, 5, 6, and 23 hours to compute for `Paris`, `Shanghai`, `L.A.` and `London`, respectively, when $k = 16$.

## C.3 THE ROLES OF GRAPH TOPOLOGY AND SPATIAL FEATURES

**The importance of structural and spatial information.** Our goal is to use a label generation approach that incorporates long-range dependency via information from both node features and graph structures. Although the local eccentricity is based on a weighted shortest-path strategy with road length being the edge weight, it does not only capture a structure perspective. This is because road length naturally relates to geographical locations, road type, land use, etc., which will link the node labels to these node features. Since the edge features are removed for the transductive task, the model will not be able to directly infer the original weighted shortest-path values. Moreover, while we can control the range of the task by changing the approximation hop $k$, this knowledge is hidden from the model. Therefore, the model needs to explore the distant neighborhood and capture both structural and spatial information to effectively infer the long-range information.

**Problem of uninformative graph structures on labels.** One catastrophic problem we encountered during the experiment is that, when using the exact eccentricity or a relatively larger $k$ for node labels, the signal becomes nearly independent of graph structures. As illustrated in Figure 7, we can observe a clear correlation between the node labels and their spatial coordinates, since nodes around the city center would generally have lower eccentricity values due to the 2D grid-like topology. This means that the spatial coordinates alone are sufficient for modeling $\varepsilon(v)$, hence weakening the importance of graph structure for the prediction task. Similarly, some other labeling approaches would also lead to the same problem, such as using a single node as an anchor, then regarding its shortest path distance to other nodes as the node label.

**Empirical validation.** In Section 3, we use an MLP on all four city networks that only uses node features such as geographical coordinates, land use, etc. The result in Figure 3 shows a significant performance gap between MLP and other graph models, indicating that using spatial information alone is insufficient for our task. To further show the sensitivity of the baseline models to geographic coordinates, we test two GNNs (GCN and GraphSAGE) and two GTs (Exphormer and SGFormer) on Paris and Shanghai, with ***coordinates masked*** in node features. At the same time, we also test MLP on these two cities with ***coordinates only***, and summarize the results in Table 5 below. Compared to the original results with all spatial features, we can observe a slight performance drop across both GNNs and GTs after removing the geographical coordinates, while for MLP, the results indicate that geographical coordinates alone are not sufficient for modelling our long-range signal.

Table 5: Baseline results on `Paris` and `Shanghai` with all features vs. coordinates only (MLP) and coordinates masked (GNNs and GTs).

| Model | Paris | | Shanghai | |
| --- | --- | --- | --- | --- |
| | all features | coords. only / masked | all features | coords. only / masked |
| MLP | $25.5 \pm 0.4$ | $12.5 \pm 0.5$ | $28.4 \pm 0.6$ | $15.2 \pm 0.7$ |
| GCN | $53.2 \pm 0.3$ | $51.4 \pm 0.4$ | $62.1 \pm 0.2$ | $61.3 \pm 0.4$ |
| GraphSAGE | $54.6 \pm 0.2$ | $52.3 \pm 0.3$ | $68.3 \pm 0.5$ | $66.5 \pm 0.4$ |
| Exphormer | $55.1 \pm 0.8$ | $53.5 \pm 0.4$ | $70.2 \pm 0.4$ | $67.4 \pm 0.5$ |
| SGFormer | $52.0 \pm 0.8$ | $51.3 \pm 0.7$ | $64.1 \pm 0.3$ | $62.8 \pm 0.4$ |

**Discussion on LRGB.** Finally, we would like to point out that this problem also exists in `LRGB`, as illustrated by an example graph from `PascalVOC-SP` in Figure 8. However, while the learning task in `PascalVOC-SP` resembles a spatial segmentation problem on the coordinates, it still requires the model to handle graph structural information for prediction due to the inductive task nature.

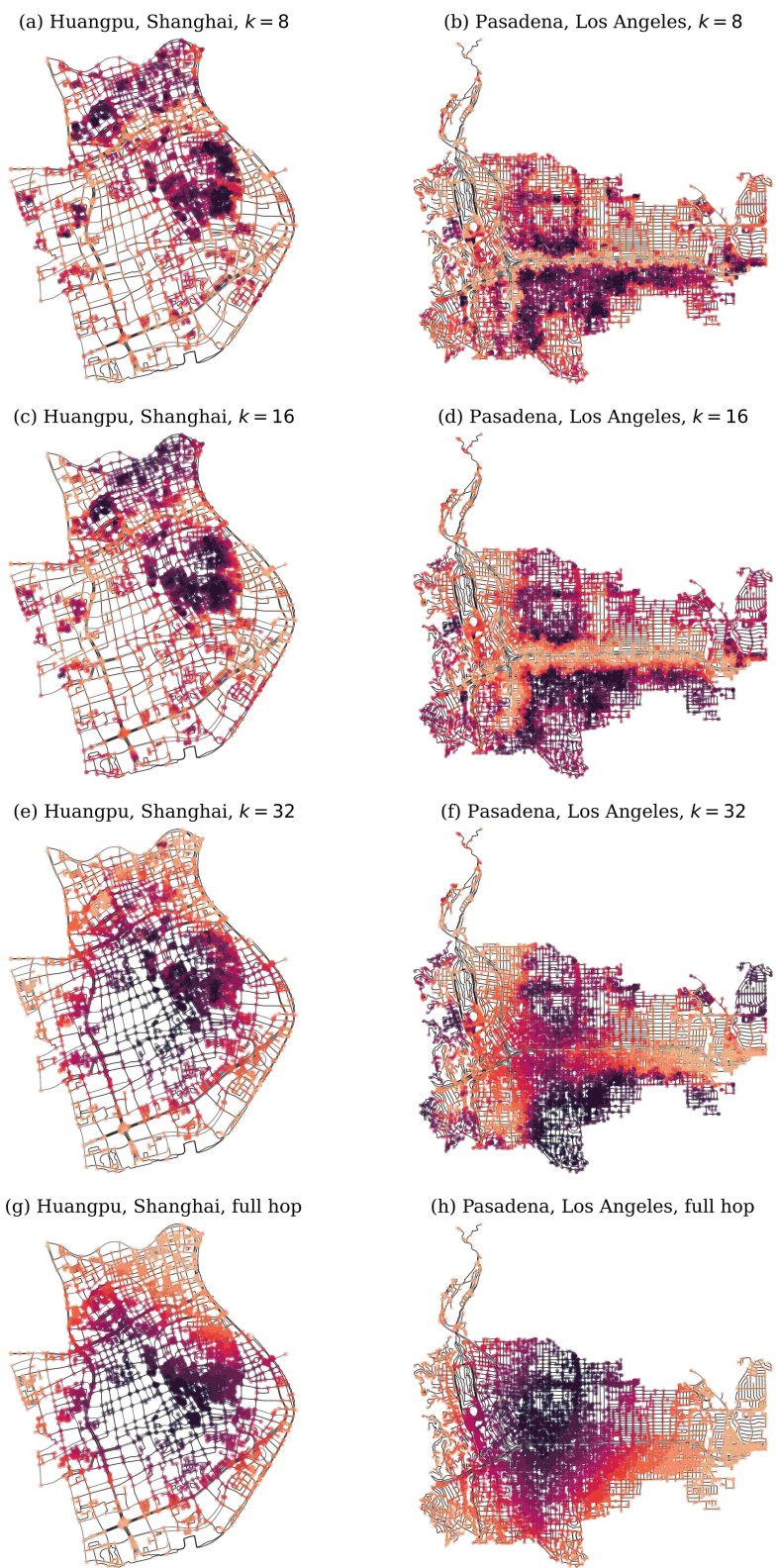

Figure 7: Visualization of local eccentricity $\hat{\varepsilon}_k(v)$ at $k = [8, 16, 32]$ and full-hop (exact eccentricity) on two sub-regions. We can observe that in the last two cases (g) and (h), node labels become highly correlated with geographic coordinates and are hence less dependent on the graph structure.

Original Image

PascalVOC-SP Network

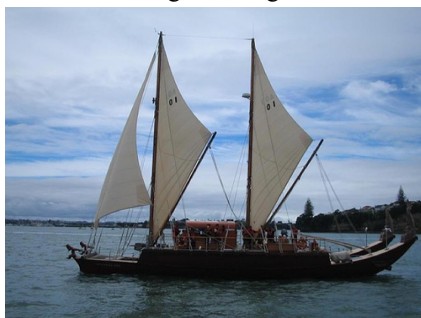
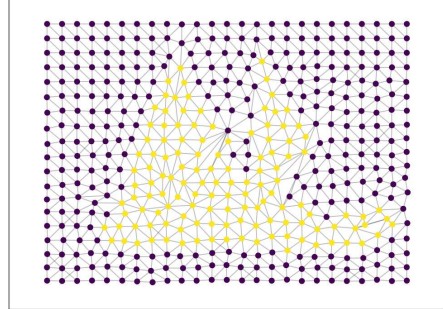

Figure 8: An example graph in `PascalVOC-SP` from `LRGB` (Dwivedi et al., 2023), where nodes are represented by super-pixels and their labels are annotated by semantics. Although spatial coordinates are highly correlated with node labels, the inductive nature makes the task still graph-dependent.

### C.4 Choice of the Long-range Level K

As explained in Section 2.2, the choice of $k$ is critical in designing the long-range signal. On one side, $k$ should be sufficiently large to distinguish our setting from short-range datasets (e.g., social graphs with typical message passing around 4 hops). To further elaborate on this, we follow the same experiment setups in Section 3 and test GCN on `Paris` using $k = [8, 16, 32]$ at num_layers=$[2, 4, 8, 16, 32]$, where the results are reported in Figure 9 on the right.

We can observe that, at $k = 8$, the model's performance quickly saturates as expected when num_layers reaches 8. While at $k = 16$ and 32, the performance starts to plateau at a much deeper depth of num_layers=16. As such, 8 does not seem to be an optimal choice for $k$, since there is still room to "extend" the range of the signal.

On the other hand, $k$ should also remain small enough that it does not lead to the aforementioned problem of uninformative graph structure during modelling. As visualized in Figure 7, the underlying signal, given its local nature, becomes smoother as $k$ increases from 8 to 32. At the same time, we can also see a increasing

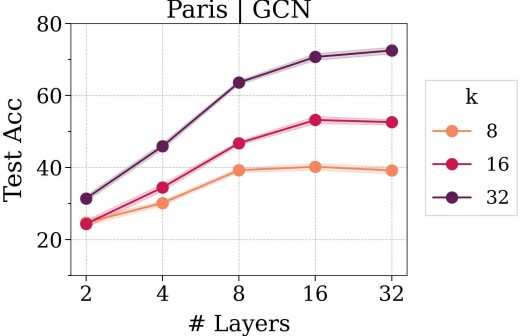

Figure 9: Results of GCN at different numbers of layers on Paris with $k = [8, 16, 32]$ used for labels.

correlation between signal and the spatial coordinates. In the extreme case when $k$ equals the network diameter, $\hat{\varepsilon}_k(v)$ becomes the exact eccentricity $\hat{\varepsilon}(v)$, and the classification task resembles a "segmentation" problem on a 2D plane. This also explains the marginal performance gain from num_layers=16 to 32 at $k = 32$, as the signal becomes overly smoothed that information based on 16-hop neighborhood is sufficient to have a good "guess" of the signal at hop 32.

At the same time, for benchmarking purposes, we wish to test GNNs and GTs with enough model depth such that they can reach the source of the signal. However, we found that when training GTs on our large city networks, due to the quadratic complexity in their global attention mechanism, they often suffer from Out-Of-Memory errors (`NVIDIA-L40@48GB`) when num_layers is large.

Therefore, we decided to adopt $k = 16$ in our final dataset, which we believe achieves a balance between being sufficiently long-ranged and remaining practical for computation. As discussed earlier in Section 1 and Section 2.2, we believe `City-Networks` brings a new and important challenge to the literature, which calls for scalable architectures that can effectively handle long-range dependency.

# D EXPERIMENTAL DETAILS

In this section, we first provide training details in our experiments, and then present an ablation study that supports our main claim on the long-range dependency. Lastly, we summarize the best baseline results on our dataset and discuss potential limitations of our experimental settings.

## D.1 BASELINES AND DATASETS FOR COMPARISON

**Baselines.** We consider four common GNNs: GCN (Kipf & Welling, 2017), GraphSAGE (Hamilton et al., 2017), GAT (Veličković et al., 2018), and GCNII (Chen et al., 2020) for GNNs; and three Graph Transformers: GraphGPS (Rampášek et al., 2022), Exphormer (Shirzad et al., 2023), and SGFormer (Wu et al., 2023) as the baseline models in our experiments.

**Datasets.** We compare the baseline results on `City-Networks` to the following datasets that are homophilic, heterophilic, large-scale, and long-range dependent.

- `Cora` (Yang et al., 2016a) is a well-known homophilic citation network, with nodes representing documents and edges representing citation links. The features of the nodes are represented as bag-of-words that captures the content of the documents, and the goal is to predict the academic topic of each paper.
- `ogbn-arxiv` (Hu et al., 2021) is a large-scale citation network of $169k$ computer science papers on arXiv that were indexed by the Microsoft academic graph, in which nodes represent papers and the directed edges indicate the citations. In particular, each node has an 128-dimensional feature vector derived from the word embeddings of titles and abstracts in the underlying papers. The task is to identify the primary category of each arXiv paper, that is, to classify each node into one of the 40 classes.
- `Amazon-Ratings` (Platonov et al., 2023) is a heterophilic network that models Amazon product co-purchasing information, where nodes represent products and edges represent frequently co-purchased items. The goal is to predict average product ratings from five classes, with node features being the fastText embeddings of product descriptions.
- `PascalVOC-SP` is an inductive dataset from `LRGB` (Dwivedi et al., 2023), which contains graphs derived from images. In particular, the nodes represent super-pixels and edges represent their boundaries. The labels are derived based on semantics, which makes the task similar to image segmentation.

As mentioned in Section 3, we closely follow the latest GNN tuning technique from Luo et al. (2024), which considers residual connection, batch normalization, dropout, etc., and use their code base for training classical GNNs and GTs on most datasets, except for `PascalVOC-SP`, where we adopt the hyperparameters reported by Tönshoff et al. (2024) and Shirzad et al. (2023).

## D.2 EXPERIMENTAL SETUPS

As discussed earlier in Section 3, we consider transductive node classification with train/validation/test splits of $10\%/10\%/80\%$ on all four graphs in `City-Networks`.

**Evaluation protocols.** Since our goal is to investigate the presence of long-range dependency, we train each model at num_layers=[2, 4, 8, 16] while fixing hidden_size=128, and then check if the model's performance will positively correlate with num_layers. All cases are repeated 5 times, and we present their means and standard deviations. In the ablation studies below, we also investigate different choices of hidden_size in [16, 32, 64, 128] following this strategy, which *not only acts as hyperparameter tuning, but also shows the robustness of our conclusion to different choices of model hyperparameters*.

**Hyperparameters** We run each model for $30k$ maximum epochs using AdamW (Loshchilov & Hutter, 2019) optimizer for sufficient training, during which we record the validation accuracy every 100 epochs and save the model at the best validation epoch for final testing. Meanwhile, we closely follow the latest GNN tuning technique from Luo et al. (2024), and our hyperparameter search space is summarized in Table 6. Note that for GTs, we do not apply positional encoding due to its

impractical computation on our large graphs (which is one of the challenges of applying GTs on large graphs), and use GCN as their internal MPNNs.

Table 6: Summary of hyperparameters and their search space.

| Type | Hyper-parameter | Search range | Default |
|---|---|---|---|
| Model | num_layers | [2, 4, 8, 16] | 16 |
| | hidden_size | [16, 32, 64, 128] | 128 |
| | pre_linear_layer | [0, 1, 2] | 0 |
| | post_linear_layer | [0, 1, 2] | 2 |
| | residual | [True, False] | True |
| Train | learning rate | $[10^{-4}, 5 \times 10^{-4}, 10^{-3}]$ | $10^{-3}$ |
| | dropout | [0, 0.2, 0.5, 0.7] | 0.2 |
| | weight decay | $[0, 10^{-5}, 5 \times 10^{-5}, 10^{-4}]$ | $10^{-5}$ |
| | normalization | [None, BatchNorm, LayerNorm] | BatchNorm |

## D.3 ABLATION STUDIES

**Results under different hidden channel sizes.** To further support our main findings, we test two classical GNN baselines: GCN (Kipf & Welling, 2017) and GraphSAGE (Hamilton et al., 2017); and two GT baselines: Exphormer (Shirzad et al., 2023) and SGFormer (Wu et al., 2023) under different hidden_size=[16, 32, 64, 128] on `Paris`, `Shanghai`, and `L.A.`. The results are presented in Figure 10, where we can observe a consistent pattern with our main findings in Figure 3—the baselines' performance improves substantially when increasing the number of layers from 2 to 16.

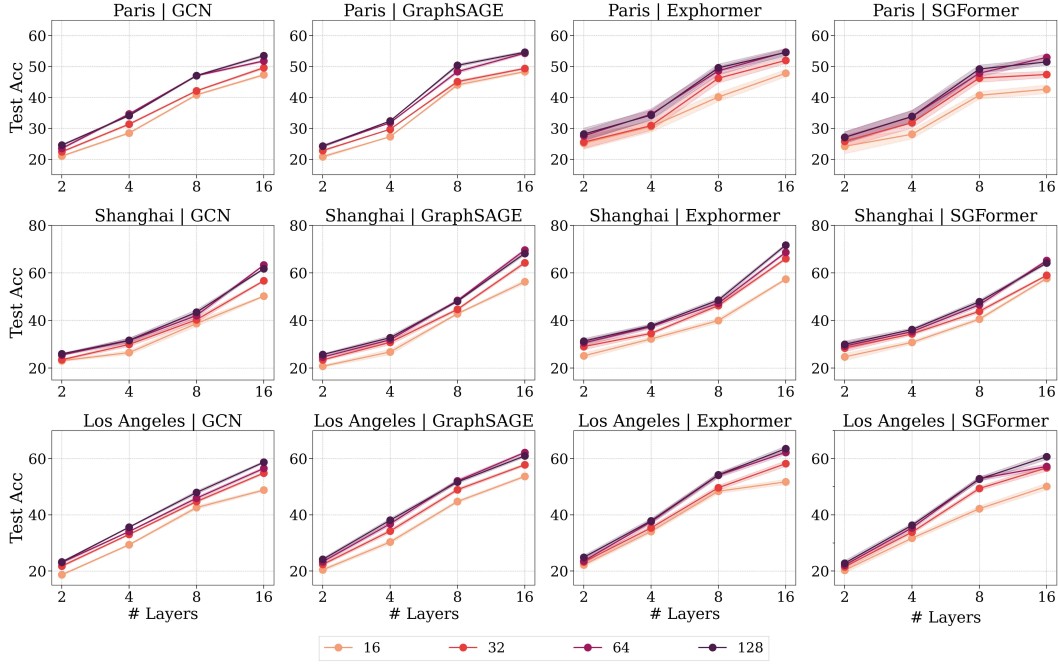

Figure 10: Ablation results under hidden_size=[16, 32, 64, 128] and num_layers=[2, 4, 8, 16] on Paris, Shanghai, and Los Angeles. The patterns are all consistent with our findings in Section 3.

Notably, all baselines generally achieve superior performance with a larger hidden_size. However, such differences are often negligible compared to the increasing trend in model depths. With that being said, in the next section, we proceed to investigate scenarios when the model size is fixed.

**Results at fix model depth with various sampling hops.** As discussed earlier, one limitation of the main results on Figure 3 is that the observed performance gains may be attributed to the

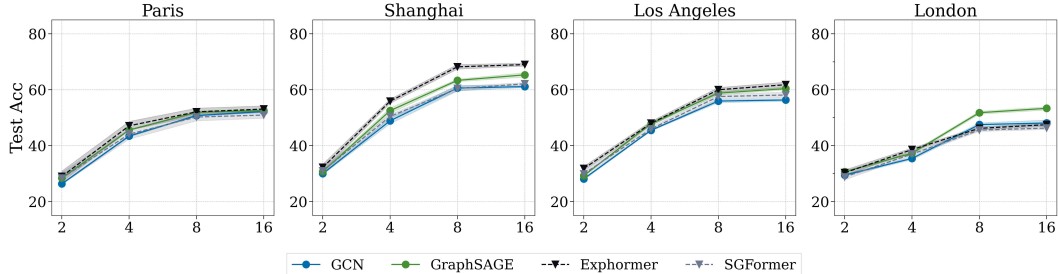

Figure 11: Results for fixing number of layers $L = 16$ and setting numbers of hops $H \in [2, 4, 8, 16]$.

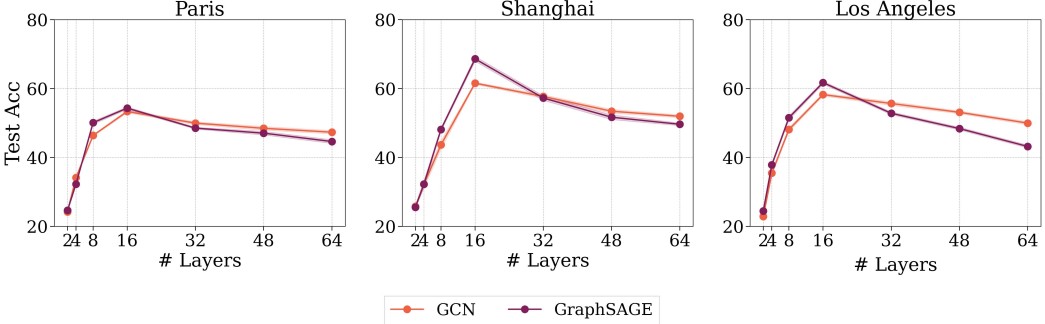

Figure 12: Results for models at deeper depths $L \in [32, 48, 64]$ beyond the ground-truth range of 16 hops. For illustration purposes, we also show results at model depths of $[2, 4, 8, 16]$.

increasing model parameters as the number of layers $L$ grows. To address this, we further investigate the scenario where $L$ is fixed at 16 (i.e., keeping the model size constant), and adopt an $H$-hop neighborhood sampling method introduced in GraphSAGE (Hamilton et al., 2017) that refrains the model from seeing beyond hop $H$. Note that the same method is also widely used in the literature for training models on large graphs.

Concretely, our strategy is implemented with `NeighborLoader` from PyG, which recursively selects $[N_1, N_2, \ldots, N_H]$ neighbors from a node's 1st, 2nd, ..., $H$-th hop neighborhood. Given the grid-like structure of our city networks, the size of the neighborhood does not explode with $H$, and we empirically found that the average size of a 16-hop ego-network is typically around $1k$ nodes, which remains manageable for most GPU devices. Therefore, we sample all nodes inside the $H$-hop neighborhood, and use a `batch_size` of $20k$ (i.e., $20k$ seed nodes) for training and testing.

The results are presented in Figure 11, where we can observe that every baseline, at a fixed model architecture, shows a consistent upward trend on all four city networks. These results indicate that long-range dependency, rather than the parameter size, is the primary factor that contributes to the improvement of the model's performance.

One may notice the trend from $H = 8$ to $H = 16$ remains relatively flat. We attribute this phenomenon to the overlapped subgraphs sampled from `NeighborLoader`, where the ego-networks of different seed nodes often share common supporting nodes. Consequently, if the number of layers $L$ exceeds $H$, the model can capture information from more distant hops beyond $H$, rather than being limited to a maximum of $H$ as expected. This effect is further amplified when using a large batch size (i.e., a large number of seed nodes) or a large number of hops (i.e. a large neighborhood size), as it increases the chance of overlapping ego-networks. As a result, the performance remains similar between these two settings. Note that this phenomenon is not a contradiction to our main claims, but rather a limitation, as we can not create perfect 8-hop subgraphs.

**Deeper depths beyond the ground-truth range.** While adopting a fixed ground-truth range prevents our long-range task from falling into an unbounded "the longer the communication path, the better the model performance" setting, we additionally benchmark GNNs with depths much deeper than the ground-truth task range (16th hop) here for a more complete analysis of the model's behavior.

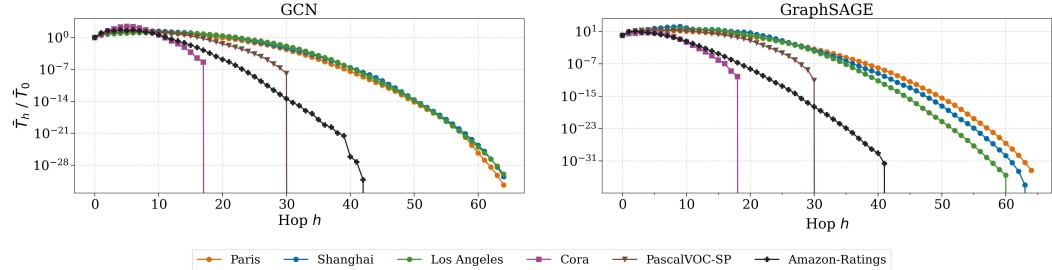

Figure 13: Normalized average total influence $\bar{T}_h/\bar{T}_0$ averaged across nodes, where the underlying models are trained with 64 layers—much deeper than the ground-truth range of the task.

Table 7: Average size of the influence-weighted receptive field $R$ across datasets on GCN and GraphSAGE, where both models are trained with 64 layers with residual connections and batch norm.

| Model | Paris | Shanghai | L.A. | Cora | Amazon | PascalVOC |
|---|---|---|---|---|---|---|
| GCN | 8.61 | 10.21 | 9.64 | 5.17 | 6.51 | 7.06 |
| GraphSAGE | 8.34 | 10.72 | 10.54 | 3.74 | 2.78 | 7.62 |

In particular, we test GCN and GraphSAGE with depth=[32, 48, 64] on Paris, Shanghai, and L.A. under the same experimental settings (GTs are not tested here due to OOM at these depths), where results are presented in Figure 12. As expected, we can observe that both GNNs, even with residual connections and batch norm enabled, start to suffer from over-smoothing when model depth exceeds the ground-truth range of 16.

In addition, we also show the results of our per-hop influence measurement and $R$ in Figure 13 and Table 7, respectively, where we compare the behaviors of these two deep GNNs on our city networks to those on the other common graph datasets. The results reveal a consistent trend with our findings in Section 4 that the influence scores decay at a much slower rate on our city networks compared to those on the existing graph datasets in the literature. It is also worth noting that the current model depth of 64 exceeds the diameters of those common graph benchmarks, which leads to the "cut-off" pattern in Figure 12.

### D.4 RESULTS ON RINGTRANSFER

The `RingTransfer` (Bodnar et al., 2021) experiment is used for testing long-range dependency in GNNs under inductive settings with small ring-like graphs. In this section, we will analyze the behaviors of different graph models on this synthetic task using our influence metric.

**Task description.** Each graph in the Ring-Transfer dataset is a ring of $N$ nodes with only two nodes marked: the source node and the target node, which are placed at opposite sides of the ring (i.e., at a distance of the diameter $N/2$). All nodes on the ring will have a constant feature vector except for the source node, which has a one-hot encoding of its label. The task is to train a model such that the target node's representation predicts the source's label, which requires the model to propagate long-range information from the source node to the target node.

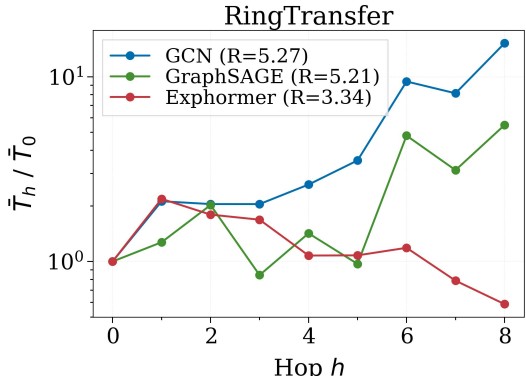

Figure 14: Influence scores on `RingTransfer`.

**Setups.** We followed the same setups by Bodnar et al. (2021) and adopted k=16 (i.e., a task range of 8 hops) with a model depth of 8 in our setting, as we observed a similar phenomenon in their paper that GNNs start to deteriorate after this point. Meanwhile, since the diameter of rings matches the true task range of $N/2$ in

`RingTransfer`, there will be no gradient w.r.t. nodes from hops $> N/2$. Therefore, we test models at depth up to the ground-truth range in this setting. In particular, we evaluate GCN, GraphSAGE, and Exphormer with training/validation/test splits of $5k/1k/1k$ graphs, where all models can achieve $100\%$ accuracy on the testing set. We then apply our measurement on the target node only for each graph, and then report the average over the testing set.

**Results and discussion.** The results for $R$ and per-hop influence are summarized in Figure 14. As expected, we can observe a strong influence at hop 8 for GCN and GraphSAGE; while for Exphormer, as it adopts global virtual node and expander graph operations in its attention mechanism, the source node can be effectively reached within a single hop. Therefore, we observe a higher influence on the first few hops compared to more distant hops since the underlying graph has been modified. In addition, we have also tested models with deeper depth of 16 layers, where the result shows a similar pattern to that in Figure 14, except for the 0 influence after hop 8 (as explained above).

While we do observe different influence patterns due to different model designs, we'd also like to point out that this is because RingTransfer is too simplistic and relies solely on long-range interactions, such that simple operations (e.g., rewiring, virtual node, etc.) will convert it into a short-range task.

### D.5    MORE RESULTS ON HETEROPHILIC DATASETS

As pointed out by Arnaiz-Rodriguez & Errica (2025), the heterophilic datasets are often misused in the graph machine learning literature to evaluate long-range interaction (Arnaiz-Rodríguez et al., 2022; Maskey et al., 2023; Giraldo et al., 2023; Singh et al., 2025). However, there is no empirical study that compares the range of tasks between homophilic and heterophilic graphs.

In Section 3 and 4, we have tested baselines and quantified their long-range influence on a representative heterophilic dataset, `Amazon-Ratings`. To further support our findings that heterophily does not empirically imply long-rangeness, we proceed to show more results on the following heterophilic datasets that are commonly used in the literature: `Roman-Empire` (Platonov et al., 2023), `Wisconsin`, `Texas`, and `Cornell` (Pei et al., 2020). The results for per-hop influence and $R$ are presented in Figure 15 and Table 8, respectively, where we can observe from these two diagnostics that heterophilic datasets generally exhibit a short-range pattern compared to our city networks across different models, which is consistent with the findings in our paper.

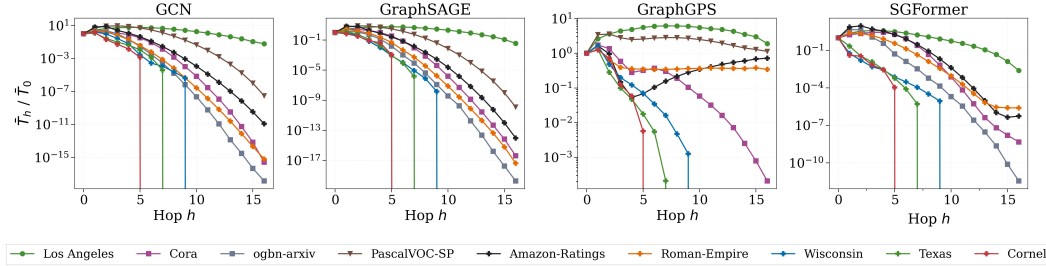

Figure 15: Normalized average total influence $\bar{T}_h/\bar{T}_0$ averaged across nodes at different hops on heterophilic datasets, where results on other datasets are also presented for comparison purposes.

Table 8: Average size of the influence-weighted receptive field $R$ across heterophilic datasets and models with 16 layers, where results on other datasets are also presented for comparison purposes.

| Model | L.A. | Cora | arxiv | PascalVOC | Amazon | Roman | Wisconsin | Texas | Cornell |
|---|---|---|---|---|---|---|---|---|---|
| GCN | 5.36 | 2.56 | 1.34 | 3.38 | 1.92 | 2.04 | 1.28 | 1.11 | 1.54 |
| GraphSAGE | 5.44 | 2.37 | 1.40 | 2.99 | 1.80 | 1.89 | 0.97 | 0.89 | 0.96 |
| GraphGPS | 7.92 | 2.65 | OOM | 7.14 | 6.86 | 6.37 | 1.03 | 0.77 | 0.98 |
| SGFormer | 4.03 | 3.21 | 1.21 | NA | 2.46 | 2.33 | 0.14 | 0.24 | 0.17 |

Note that `Wisconsin`, `Texas`, and `Cornell` are small heterophilic graphs (around 200 nodes) and all have a small diameter of 8, which explains their extreme short-range behaviors compared to other heterophilic datasets like `Amazon-Ratings` and `Roman-Empire`.

## E  COMPUTATIONAL COMPLEXITY

The computation of $R$ requires calculating $T_h(v)$ for $v \in V$ at $h \in [0, 1, ..., H]$, in which the dominant cost stems from computing the Jacobian matrix. Since the model's gradient at node $v$ will be zero for nodes beyond its $H^{th}$ hop, we only need to compute the Jacobian within each node's $H$-hop neighborhood. This leads to a computational cost of $\mathcal{O}(N\bar{N}_H)$, with $\bar{N}_H$ being the average size of $H$-hop ego-networks on $\mathcal{G}$:

$$\bar{N}_H = \frac{1}{N} \sum_{v \in V} |\mathcal{N}_H(v)|, \quad \mathcal{N}_H(v) = \{u \in V \mid d(v, u) \leq H\}, \tag{13}$$

where $d(v, u)$ is the length of shortest path from node $v$ to node $u$. Given the large scale of our city networks, we employ a stochastic approximation that samples $10k$ nodes to further reduce the computational cost. For reference, the calculation finishes under 30 minutes for all baselines on all four `City-Networks` using a single `NVIDIA RTX 3090` GPU with 48 `AMD Ryzen 3960X`CPU cores.

## F  LIMITATIONS AND FUTURE WORK

***Total Influence* under model bias.**  Since our measurement is based on the gradient of the underlying model, its behavior will be naturally influenced by the bias in the architecture. For example, as described in Appendix D.3, when the models' depth (num_layers=64) is much deeper than the known task depth ($k = 16$), we observe that $R$s are generally larger than those from models with 16 layers in our city networks. Meanwhile, the per-hop influence also suggests that deeper models leverage information from distant hops beyond the ground truth range $k$ (which makes sense since the model has no information about $k$, and node features beyond the $k$th hop may contain useful information for prediction). Therefore, the biased approximation of the ground-truth function will inevitably lead to a biased estimation of the underlying task's range, even though the measurement remains a faithful description of how the model utilizes long-range information.

However, our conclusions regarding long-rangeness across datasets are still valid, since each model is validated across different datasets under the same depth in our analysis, where all models show clear long–range patterns on our city-networks compared to those on the existing benchmarks at depth=16 (Section 4) and depth=64 (Appendix D.3).

***Total Influence* on large dense graphs.**  During our experiment, we found that on large dense graphs, especially the one with small diameters such as `ogbn-arxiv` ($|V| = 169k$, $diam = 25$), it is difficult to compute our *Total Influence* measurement, as $\bar{N}_H$ in equation 13 quickly converges to $N$ as $H$ increases. In this case, if the model also happens to have complex architectures (e.g., GraphGPS), the computation of $R$ and $T_h$ will therefore become impractical. Meanwhile, we also want to point out that this is a common challenge for all Jacobian-based analysis in the literature (Xu et al., 2018; Gasteiger et al., 2022), and it is an open question for future works to explore more effective methods for measuring long-range dependency on dense graphs.

**Inductive setting.**  In the current work, we focus on testing long-range signals under transductive settings on large-scale graphs, which is largely underexplored in the literature. However, we believe it is also possible to extend our method to inductive settings by sampling a set of cities' road networks via OpenStreetMap (or similar geographic graph sources), and then defining a graph-level classification or regression task on them, e.g., urban (graph) morphology classification. We leave this direction for future explorations.

## G  PROOFS AND JUSTIFICATIONS IN SECTION 5

### G.1  DETAILS OF LINEARIZATION

A Graph Convolutional Network (GCN) layer with input features $\boldsymbol{X}^{(l)}$, learnable weights $\boldsymbol{W}^{(l)}$ for layer $l$, and non-linear pointwise activation function $\sigma$ is defined as:

$$\boldsymbol{X}^{(l+1)} = \sigma \left( \tilde{\boldsymbol{S}}_{adj} \boldsymbol{X}^{(l)} \boldsymbol{W}^{(l)} \right). \tag{14}$$

As pointed out in Wu et al. (2019), under linearization (removal of non-linear activations), an $L$-layer Simple Graph Convolution (SGC) can be expressed as:

$$\boldsymbol{X}^{(L)} = (\tilde{\boldsymbol{S}}_{adj})^L \boldsymbol{X}^{(0)} (\prod_{l=0}^{L-1} \boldsymbol{W}^{(l)}), \tag{15}$$

where $(\tilde{\boldsymbol{S}}_{adj})^L$ is the $L$-th power of the normalized adjacency operator. The result follows directly from iterative application of the linearized GCN operation.

Next, it is well known that the normalized adjacency operator $\tilde{\boldsymbol{S}}_{adj}$ admits an eigendecomposition:

$$\tilde{\boldsymbol{S}}_{adj} = \tilde{\boldsymbol{U}} \boldsymbol{\Lambda} \tilde{\boldsymbol{U}}^T, \tag{16}$$

where $\boldsymbol{\Lambda} = \text{diag}(\lambda_1, \ldots, \lambda_N)$ with $\lambda_1 < \ldots < \lambda_N$ and $\tilde{\boldsymbol{U}}$ contains the corresponding eigenvectors. And, for any positive integer $k$, the power iteration is as follows:

$$(\tilde{\boldsymbol{S}}_{adj})^k = \tilde{\boldsymbol{U}} \boldsymbol{\Lambda}^k \tilde{\boldsymbol{U}}^T, \tag{17}$$

where $\boldsymbol{\Lambda}^k = \text{diag}(\lambda_1^k, \ldots, \lambda_N^k)$.

Wu et al. (2019) discuss the spectral properties of the normalized adjacency operator, which satisfies: $\lambda_N = 1$, and $|\lambda_i| < 1$ for all $i < N$. Therefore, as the number of layers approaches infinity, the layer collapse phenomenon is well-documented in the literature:

$$\lim_{l \to \infty} (\tilde{\boldsymbol{S}}_{adj})^l = \tilde{\boldsymbol{u}}_N \tilde{\boldsymbol{u}}_N^T, \tag{18}$$

where $\tilde{\boldsymbol{u}}_n$ is the eigenvector corresponding to $\lambda_N$. This is because as $l \to \infty$, $\lambda_i^l \to 0$ for all $i < N$ since $|\lambda_i| < 1$. Meanwhile, $\lambda_N^l = 1^l = 1$ for all $l$. Thus,

$$\lim_{l \to \infty} (\tilde{\boldsymbol{S}}_{adj})^l = \tilde{\boldsymbol{U}} \begin{pmatrix} 0 & & \\ & \ddots & \\ & & 1 \end{pmatrix} \tilde{\boldsymbol{U}}^T = \tilde{\boldsymbol{u}}_N \tilde{\boldsymbol{u}}_N^T. \tag{19}$$

This ultimately leads to information loss. In the limit as $k \to \infty$, the graph convolution operation collapses all node features to scalar multiples of $\tilde{\boldsymbol{u}}_n$, the entry of which at node $v$ is $\sqrt{1 + \text{degree}(v)}$, resulting in complete loss of the original feature information. In other words, the learned representations suffer from *over-smoothing*.

### G.2 PROOFS FOR THEORETICAL RESULTS IN SECTION 5

**Lemma G.1** (Eigenvalue complementarity of normalized operators). For a connected graph, the eigenvalues $\boldsymbol{S}_{adj}$ and $\boldsymbol{L}_{sym}$ exhibit the following complementarity relationship:

$$\lambda_{N+1-i}(\boldsymbol{S}_{adj}) = 1 - \lambda_i(\boldsymbol{L}_{sym}) \tag{20}$$

for all $i = 1, \ldots, N$, where $N$ is the number of nodes in the graph, and eigenvalues (for both operators) are indexed such that $\lambda_1 \le \lambda_2 \le \cdots \le \lambda_N$.

*Proof of Lemma G.1.* By definition, $\boldsymbol{L}_{sym} = \boldsymbol{I} - \boldsymbol{S}_{adj}$. Since both matrices are symmetric, they are diagonalizable with real eigenvalues. Let $\boldsymbol{u}$ be an eigenvector of $\boldsymbol{L}_{sym}$ with eigenvalue $\lambda_i(\boldsymbol{L}_{sym})$. Then:

$$\boldsymbol{L}_{sym}\boldsymbol{u} = \lambda_i(\boldsymbol{L}_{sym})\boldsymbol{u} \Rightarrow (\boldsymbol{I} - \boldsymbol{S}_{adj})\boldsymbol{u} = \lambda_i(\boldsymbol{L}_{sym})\boldsymbol{u} \Rightarrow \boldsymbol{S}_{adj}\boldsymbol{u} = (1 - \lambda_i(\boldsymbol{L}_{sym}))\boldsymbol{u}. \tag{21}$$

Therefore, $\boldsymbol{u}$ is also an eigenvector of $\boldsymbol{S}_{adj}$ with eigenvalue $1 - \lambda_i(\boldsymbol{L}_{sym})$. Since the eigenvalues are ordered in ascending order for $\boldsymbol{L}_{sym}$ and the transformation $1 - \lambda_i$ reverses this ordering, we have $\lambda_{N+1-i}(\boldsymbol{S}_{adj}) = 1 - \lambda_i(\boldsymbol{L}_{sym})$. $\square$

This complementarity implies that when the normalized algebraic connectivity $\lambda_2(\boldsymbol{L}_{sym})$ is small, the second largest positive eigenvalue of $\boldsymbol{S}_{adj}$ must be close to 1. However, in our context, we are interested in the algebraic connectivity of the normalized graph Laplacian that would correspond to the normalized adjacency operator, $\tilde{\boldsymbol{S}}_{adj}$, introduced in Section 5.1, instead of that of $\boldsymbol{S}_{adj}$, motivating the following:

**Proposition G.2** (Self-loops decrease algebraic connectivity of the original graph, from Section 5.2)**.**
Assume a connected graph $\mathcal{G}$ with more than two nodes. For all $\gamma > 0$,

$$\lambda_{N-1}\left(\boldsymbol{S}_{adj}\right) = \lambda_{N-1}\left(\boldsymbol{D}^{-\frac{1}{2}}\boldsymbol{A}\boldsymbol{D}^{-\frac{1}{2}}\right) < \lambda_{N-1}\left((\gamma\boldsymbol{I}+\boldsymbol{D})^{-\frac{1}{2}}(\gamma\boldsymbol{I}+\boldsymbol{A})(\gamma\boldsymbol{I}+\boldsymbol{D})^{-\frac{1}{2}}\right). \quad (22)$$

*Proof of Proposition 5.1 (same as Proposition G.2).* Let $\mathcal{G}'$ be the graph $\mathcal{G}$ with self-loops added, each having weight $\gamma > 0$. If $\mathcal{G}$ already contains self-loops, their weights are increased by $\gamma$. We denote the vertex set of $\mathcal{G}'$ as $V$, with the obvious correspondence to the vertices of $\mathcal{G}$. Then $\boldsymbol{L}_{sym}^{\mathcal{G}} = \boldsymbol{I} - \boldsymbol{D}^{-\frac{1}{2}}\boldsymbol{A}\boldsymbol{D}^{-\frac{1}{2}}$ and $\boldsymbol{L}_{sym}^{\mathcal{G}'} = \boldsymbol{I} - (\gamma\boldsymbol{I}+\boldsymbol{D})^{-\frac{1}{2}}(\gamma\boldsymbol{I}+\boldsymbol{A})(\gamma\boldsymbol{I}+\boldsymbol{D})^{-\frac{1}{2}}$, so proving

$$\lambda_2(\boldsymbol{L}_{sym}^{\mathcal{G}}) > \lambda_2(\boldsymbol{L}_{sym}^{\mathcal{G}'}). \quad (23)$$

will prove the proposition. Note that in practice we care about the case where $\gamma = 1$. We proceed as follows: we take the eigenfunction on $\mathcal{G}$ corresponding to $\lambda_{\mathcal{G}}$, lift it to $\mathcal{G}'$, and show that this yields an upper bound. Using the variational characterisation of eigenvalues in (Chung, 1997, Eq. 1.13) and the fact that edge weights $w_{\mathcal{G}}(u, v) = w_{\mathcal{G}'}(u, v)$ if $u \neq v$, and the degrees of vertices $d_v^{\mathcal{G}'} = \gamma + d_v^{\mathcal{G}}$:

$$\lambda_2(\boldsymbol{L}_{sym}^{\mathcal{G}}) = \inf_{f \, : \, \sum_{x \in V} f(x) d_x^{\mathcal{G}} = 0} \frac{\sum_{x \sim y}(f(x) - f(y))^2 w(x, y)}{\sum_{x \in V} f(x)^2 d_x^{\mathcal{G}}} \quad (24)$$

Pick such an $f$ attaining this infimum (i.e., the eigenvector of $\boldsymbol{L}_{sym}^{\mathcal{G}}$ corresponding to $\lambda_2(\boldsymbol{L}_{sym}^{\mathcal{G}})$ multiplied by $\boldsymbol{D}^{-\frac{1}{2}}$). We use this $f$ to construct a signal $g$ on $\mathcal{G}'$. Let

$$g(x) = f(x) - \frac{\gamma \sum_{x \in V} f(x)}{\sum_{x \in V} d_x^{\mathcal{G}} + \gamma}. \quad (25)$$

That is, we reduce $f$ by a constant everywhere. We have picked the constant such that

$$\sum_{x \in V} g(x) d_x^{\mathcal{G}'} = \sum_{x \in V} g(x)(d_x^{\mathcal{G}} + \gamma) = 0. \quad (26)$$

Since $g$ is simply $f$ shifted by a constant,

$$\forall x, y \in V : \quad g(x) - g(y) = f(x) - f(y). \quad (27)$$

Again, by the variational characterisation of eigenvalues in (Chung, 1997, Eq. 1.13):

$$\lambda_2(\boldsymbol{L}_{sym}^{\mathcal{G}'}) = \inf_{h \, : \, \sum_{x \in V} h(x) d_x^{\mathcal{G}'} = 0} \frac{\sum_{x \sim y}(h(x) - h(y))^2 w(x, y)}{\sum_{x \in V} h(x)^2 d_x^{\mathcal{G}'}} \quad (28)$$

$$\leq \frac{\sum_{x \sim y}(g(x) - g(y))^2 w(x, y)}{\sum_{x \in V} g(x)^2 d_x^{\mathcal{G}'}} = \frac{\sum_{x \sim y}(f(x) - f(y))^2 w(x, y)}{\sum_{x \in V} g(x)^2 d_x^{\mathcal{G}'}} \quad (29)$$

The first inequality holds because $g$ is in the set $\{h \, : \, \sum_{x \in V} h(x) d_x^{\mathcal{G}'} = 0\}$, and the infimum serves as a lower bound for the function on any element of that set. The last equality follows from applying (27).

We now show that $\sum_{x \in V} g(x)^2 d_x^{\mathcal{G}'} > \sum_{x \in V} f(x)^2 d_x^{\mathcal{G}}$. This will let us bound (29) above by (24). Expanding the definition of $g$:

$$\sum g(x)^2 d_x^{\mathcal{G}'} = \sum f(x)^2 d_x^{\mathcal{G}'} - 2 \frac{\gamma(\sum f(x) d_x^{\mathcal{G}'})(\sum f(x))}{\sum d_x^{\mathcal{G}'}} + \frac{\gamma^2(\sum f(x))^2}{\sum d_x^{\mathcal{G}'}} \quad (30)$$

Noting that $\sum f(x) d_x^{\mathcal{G}} = 0$ and $d_x^{\mathcal{G}'} = d_x^{\mathcal{G}} + \gamma$ so $\sum f(x) d_x^{\mathcal{G}'} = \gamma \sum f(x)$,

$$\sum g(x)^2 d_x^{\mathcal{G}'} - \sum f(x)^2 d_x^{\mathcal{G}} = \gamma \sum f(x)^2 - 2 \frac{\gamma^2(\sum f(x))^2}{\sum d_x^{\mathcal{G}'}} + \frac{\gamma^2(\sum f(x))^2}{\sum d_x^{\mathcal{G}'}} \quad (31)$$

$$= \gamma \sum f(x)^2 - \gamma^2 \frac{(\sum f(x))^2}{\sum d_x^{\mathcal{G}'}} \quad (32)$$

By the Cauchy-Schwarz inequality on $f(\boldsymbol{x})$ and $\boldsymbol{1}$,

$$\left(\sum f(x)\right)^2 = \left(\sum f(x) \cdot 1\right)^2 \leq \left(\sum f(x)^2\right)\left(\sum 1^2\right) = n \sum f(x)^2. \tag{33}$$

Furthermore, as $\mathcal{G}$ is connected, each node has degree of at least $1 - \forall x : d_x^{\mathcal{G}} \geq 1$. As the graph has more than two nodes, one node must have degree of at least 2. So $\sum d_x^{\mathcal{G}'} > n(1+\gamma)$, and therefore:

$$\sum g(x)^2 d_x^{\mathcal{G}'} - \sum f(x)^2 d_x^{\mathcal{G}} \geq \frac{\gamma}{n}\left(\sum f(x)\right)^2 - \gamma^2 \frac{\left(\sum f(x)\right)^2}{n(1+\gamma)} \tag{34}$$

$$= \frac{\left(\sum f(x)\right)^2}{n}\left(\gamma - \frac{\gamma^2}{1+\gamma}\right) \tag{35}$$

$$= \frac{\left(\sum f(x)\right)^2}{n}\left(\frac{\gamma}{1+\gamma}\right) \tag{36}$$

$$> 0. \tag{37}$$

Based on which we conclude that $\sum_{x \in V} g(x)^2 d_x^{\mathcal{G}'} > \sum_{x \in V} f(x)^2 d_x^{\mathcal{G}}$, and so by (24) and (29):

$$\lambda_2(\boldsymbol{L}_{sym}^{\mathcal{G}'}) \leq \frac{\sum_{x \sim y}(f(x) - f(y))^2 w(x,y)}{\sum_{x \in V} g(x)^2 d_x^{\mathcal{G}'}} \tag{38}$$

$$< \frac{\sum_{x \sim y}(f(x) - f(y))^2 w(x,y)}{\sum_{x \in V} f(x)^2 d_x^{\mathcal{G}}} = \lambda_2(\boldsymbol{L}_{sym}^{\mathcal{G}}). \tag{39}$$

Hence, $\lambda_2(\boldsymbol{L}_{sym}^{\mathcal{G}}) > \lambda_2(\boldsymbol{L}_{sym}^{\mathcal{G}'})$ and by (23), the proof is complete. $\qquad\square$

*Proof of Theorem 5.2.* Given Lemma G.1 (correspondence between the eigenvalues of $\boldsymbol{S}_{adj}$ and $\boldsymbol{L}_{sym}^{\mathcal{G}}$) and Proposition 5.1, seeing that $diam(\mathcal{G}) \geq 4$ means that the graph has more than two nodes, the result follows immediately from (Chung, 1997, Lemma 1.14). $\qquad\square$

### G.3 RATIONALE BEHIND THEOREM 5.2

The second largest eigenvalue of $\tilde{\boldsymbol{S}}_{adj}$ in terms of magnitude is either $\lambda_{N-1}$ or $\lambda_1$. We explicitly consider the case when it is $\lambda_{N-1}$. As $\lambda_{N-1}$ approaches 1, the rate of convergence to the limiting state $\tilde{\boldsymbol{u}}_N \tilde{\boldsymbol{u}}_N^T$ decreases exponentially with the number of layers. For any layer $l$, the difference from the limiting state can be expressed as:

$$||(\boldsymbol{I} - \tilde{\boldsymbol{u}}_N \tilde{\boldsymbol{u}}_N^T)(\tilde{\boldsymbol{S}}_{adj})^l|| = ||\tilde{\boldsymbol{U}}\text{diag}(\lambda_1^l, ..., \lambda_{N-1}^l, 0)\tilde{\boldsymbol{U}}^T|| = \lambda_{N-1}^l \tag{40}$$

where $|| \cdot ||$ denotes the spectral norm and the last equality follows from that the spectral norm of a diagonal matrix being equal to the largest absolute value of its diagonal entries. Thus, when $\lambda_{N-1}$ is close to 1, more layers are required to achieve the same level of convergence to the limiting state. Since $\lambda_{N-1} < 1$, taking powers of $\lambda_{N-1}$ will eventually converge to zero, but the rate of this convergence slows dramatically as $\lambda_{N-1}$ approaches 1. Hence, because graphs with a large diameter and low maximum degree have a lower algebraic connectivity due to reduced inter-component connectivity, GNNs operating on such graphs will be less susceptible to over-smoothing when processing node features.

We have focused on the case where the second largest eigenvalue by magnitude of $\tilde{\boldsymbol{S}}_{adj}$ is $\lambda_{N-1}$. The special case where it is instead $\lambda_1$ (which, in this case, must be negative) gives a different regime, where it is possible for graphs with smaller diameters to exhibit less over-smoothing than those with larger diameters. Consider, for example, a bipartite graph where over-smoothing occurs independently on each side of the partition, resulting in two distinct values depending on which partition the node is in, rather than convergence to a multiple of $\tilde{\boldsymbol{u}}_N$, where the node feature value would only depend on the degree.

Nevertheless, our analysis demonstrates that large diameters and sparse connectivity generally reduce the likelihood of over-smoothing. This insight motivates our proposal of benchmark datasets with significantly larger diameters than those currently used in the literature (see Table 1). The underlying hypothesis is that reduced likelihood of over-smoothing in high-diameter, sparse networks enables the possibility of GNNs learning representations that capture long-range dependencies when necessary.

G.4  RELATIONS TO BRAESS PARADOX AND PREVIOUS WORK

**Sparsity of the graph and spectral gap.**   Our main theoretical result in Section 5 (Theorem 5.2) suggests that the second largest eigenvalue of the normalized augmented adjacency matrix has a lower bound that is increasing in graph diameter and decreasing in maximum node degree, hence the spectral gap tends to be smaller for graphs with larger diameter and smaller maximum degree. However, this tendency may not be exact as the bound is not necessarily tight, and furthermore, a smaller maximum degree does not necessarily imply a sparser graph (although the two can be generally correlated). Therefore, our result is not in contradiction with the Braess paradox, which is an interesting observation by itself.

**Spectral gap and tendency of over-smoothing.**   Our analysis of the rate of over-smoothing is purely based on the spectral analysis that the speed of convergence to a stationary point upon repeated application of a matrix operator depends on the spectral gap: a smaller gap generally slows down convergence in this precise sense. It is worth noting that this analysis is **task-agnostic**, i.e., it does not take into account node classification as a task, and furthermore distribution of node labels. The analysis of over-smoothing by Jamadandi et al. (2024) is, however, **task-dependent**, as it specifically highlights situations when pruning edges can mitigate over-smoothing of task-beneficial signals by disconnecting nodes with different labels. Therefore, while both are meaningful, the two analyses look at over-smoothing from slightly different perspectives.

# H   MOTIVATION FOR JACOBIAN-BASED INFLUENCE SCORE

We provide additional motivation behind the proposed measurement for quantifying long-rangeness in Section 4.

## H.1   INTUITION BEHIND JACOBIAN-BASED INFLUENCE SCORE

The Jacobian has been used in analysis of node interactions in GNNs in multiple previous works (Xu et al., 2018; Gasteiger et al., 2022; Di Giovanni et al., 2023). For example, it is used in Di Giovanni et al. (2023, Theorem 4.1) to show when over-squashing happens in long-range interactions, and to show how vanishing gradients occur in very deep GNNs. Influence specifically has been used to compute a natural measure of interactions between two nodes (Xu et al., 2018). We accordingly use aggregated influence, Equation equation 2, to gauge how nodes at a distance $h$ affect the output of the GNN at a focal node, thus quantifying *long-rangeness*. By the definition of partial derivatives, we can understand the Jacobian as follows:

$$\frac{\partial \boldsymbol{H}_{vi}^{(\ell)}(\boldsymbol{X})}{\partial \boldsymbol{X}_{uj}} = \lim_{\delta \to 0} \frac{\boldsymbol{H}_{vi}^{(\ell)}(\boldsymbol{X} + \delta \boldsymbol{e}_{uj}) - \boldsymbol{H}_{vi}^{(\ell)}(\boldsymbol{X})}{\delta}, \tag{41}$$

where $\boldsymbol{X}$ is the original (unperturbed) input feature matrix, and $\delta \boldsymbol{e}_{uj}$ is an infinitesimal perturbation in the $j^{th}$ component of the feature vector at node $u$ (a standard basis vector in the node feature space). This means the more positive the Jacobian entry is, the more a positive perturbation of the features at node $u$ and component $j$ will increase the logits at node $v$ and component $i$ at the final layer. In other words, the Jacobian entry being positive or negative means that the logits are pushed up or down. Given that, after applying the softmax function, the probabilities at a given data point increase monotonically with the logits at that point, we can consider both positive and negative influences as actual influence and only focus on the absolute value (i.e., sensitivity rather than direction).

*Monotonicity of the Softmax function.*  Consider the derivative of the $i$-th softmax probability with respect to its corresponding logit:

$$\frac{\partial p_i}{\partial z_i} = \frac{\partial}{\partial z_i}\left(\frac{\exp(z_i)}{\sum_j \exp(z_j)}\right) = \frac{\exp(z_i)\sum_j \exp(z_j) - \exp(z_i)\exp(z_i)}{(\sum_j \exp(z_j))^2} = p_i(1 - p_i) \tag{42}$$

Observe that: $p_i(1 - p_i) > 0$ for $0 < p_i < 1$ and $\frac{\partial p_i}{\partial z_i} \to 0$ as $p_i \to 0$ or $p_i \to 1$. For $j \neq i$:

$$\frac{\partial p_i}{\partial z_j} = \frac{\partial}{\partial z_j}\left(\frac{\exp(z_i)}{\sum_h \exp(z_h)}\right) = -\frac{\exp(z_i)\exp(z_j)}{(\sum_h \exp(z_h))^2} = -p_i p_j \tag{43}$$

This proves that the softmax probabilities increase with their corresponding logits and decrease with other logits. ☐

The above is a well-known fact and we do not claim novelty.

## H.2 POTENTIAL LIMITATIONS: OUTPUT CANCELLATION

However, it is possible to find counterexamples in which measuring the absolute Jacobian sensitivity could be insufficient or even misleading: if a positive and negative influence always cancel each other out. We find such a situation can happen in unregularised linear models with heavy collinearity of features (Hastie et al., 2009, p.63) – indeed, this is presented as one of the motivations of ridge regression. To better understand this, we give a simple model of such cancellation:

**Proposition H.1** (Absolute Jacobian sensitivity may over-estimate influence)**.** There exists a model $h_v$, where the combined effect of changes in input variables on the output is zero (i.e., the sum of the partial derivatives is zero), while the sum of the absolute values of the individual partial derivatives is nonzero.

*Proof of Proposition H.1.* Consider a graph with three nodes $u, v, w$, where $v$ is the focal node for our calculation. Let input and output features be scalars $\mathbf{X}_u = x_u$ on $\mathbf{H}_v = h_v$ (nodewise binary classification where using a single logit as input to a sigmoid function is possible). The influence is $I(v, u) = \left| \frac{\partial h_v}{\partial x_u} \right|$, and similarly for $\mathbf{X}_w = x_w$, $I(v, w) = \left| \frac{\partial h_v}{\partial x_w} \right|$. Assume the model learns the function $h_v = x_u - x_w$ and that $x_u \approx x_w$, then following the definition in the main text:

$$T_h(v) = I(v, u) + I(v, w) = \left| \frac{\partial h_v}{\partial x_u} \right| + \left| \frac{\partial h_v}{\partial x_w} \right| = |1| + |-1| = 2, \tag{44}$$

while $h_v = 0$. In fact,

$$\frac{\partial h_v}{\partial x_u} + \frac{\partial h_v}{\partial x_w} = 1 + (-1) = 0 \tag{45}$$

holds for all $(x_u, x_w)$; no small-difference assumption on $x_u$ and $x_w$ is required.

This demonstrates that the sum of the absolute values of the Jacobian entries can be non-zero, even when the net effect on the output is zero. The key is the opposing nature of their influence, not their specific values. The example sets $x_u$ and $x_w$ as approximately equal to highlight that the net output can be small (or changes to it can be small) while the individual influences are significant. ☐

Although this is a valid concern, we next show that for Message Passing Neural Networks (MPNNs), at least at initialization, such cancellation does not happen.

**Definition H.2** (Message-Passing Neural Network layer)**.** For a MPNN layer $l$, the node feature update for $v$ is given by: $\mathbf{X}_v^{(l+1)} = \phi \left( \mathbf{X}_v^{(l)}, \bigoplus_{u \in \mathcal{N}(v)} \psi \left( \mathbf{X}_v^{(l)}, \mathbf{X}_u^{(l)} \right) \right)$, where $\psi$ is a message function, responsible for computing interactions between neighboring nodes, $\bigoplus$ is a permutation-invariant aggregation function, such as summation, mean, or max, $\phi$ is an update function that integrates aggregated information into the node representation.

**Definition H.3** (Smooth Hypersurface in $\mathbb{R}^D$)**.** A hypersurface in $\mathbb{R}^D$ is a subset defined locally as the zero set of a continuously differentiable function $f : \mathbb{R}^D \to \mathbb{R}$ such that the gradient $\nabla f$ is nonzero at almost every point where $f = 0$. That is, if we have an equation of the form $f(\boldsymbol{\theta}) = 0$, where $\boldsymbol{\theta}$ is a vector of parameters, and if $\nabla f(\boldsymbol{\theta}) \neq 0$ generically, then $f = 0$ defines a (locally) $(D-1)$-dimensional manifold, which is a hypersurface.

**Theorem H.4** (Measure-zero of exact cancellation at MPNN initialization)**.** Consider an MPNN where the functions $\psi$ and $\phi$ are parameterized by $\boldsymbol{\theta}$ and differentiable, typically modeled as MultiLayer Perceptrons (MLPs). Suppose the parameters $\boldsymbol{\theta}$ are drawn from a probability distribution that is absolutely continuous with respect to the Lebesgue measure. Then, the set of parameter configurations for which exact cancellation of Jacobian contributions occurs has Lebesgue measure zero.

*Proof of Theorem H.4.* For exact cancellation to hold assuming $\bigoplus = \sum$, the following sum must be identically zero while the individual terms remain non-zero. Since $\psi$ and $\phi$ are differentiable and parameterized by $\boldsymbol{\theta}$, each Jacobian term is a smooth function of $\boldsymbol{\theta}$. The equation:

$$f(\boldsymbol{\theta}) = \sum_{u \in \mathcal{N}(v)} \frac{\partial \psi(\boldsymbol{X}_v^{(l)}, \boldsymbol{X}_u^{(l)})}{\partial \boldsymbol{X}_u^{(l)}} = 0, \tag{46}$$

defines a level set of smooth functions, a hypersurface (or a set of lower-dimensional submanifolds) in parameter space. The solution set of a nontrivial smooth function has Lebesgue measure zero unless it is identically zero across all $\boldsymbol{\theta}$, which is not the case here. Furthermore, since $\boldsymbol{\theta}$ is drawn from an absolutely continuous distribution (such as Gaussian or uniform), the probability of exactly selecting a parameter that lies on this hypersurface is zero at initialization. Thus, exact cancellation of Jacobian contributions is an event of measure zero in the space of our idealized single-layer MPNN: this can naturally be extended to multiple layers. Note that in this proof we have assumed the Jacobian sum is not identically zero by construction, unlike in Proposition H.1 where the function was explicitly constructed to ensure cancellation. $\square$

Lastly, it is worth noting an alternative perspective on interpreting influence measures derived from the sum of absolute Jacobian entries, particularly when considering potential cancellation effects as demonstrated in Proposition H.1. One can indeed argue that the measure's utility lies precisely in its capacity to quantify the magnitude of sensitivity to inputs from individual distant nodes or pathways, irrespective of whether these influences ultimately negate one another in contributing to the final output. From this stance, focusing on the sum of absolute values reveals the underlying structure and strength of potential long-range dependencies (the information channels themselves) that might be obscured by observing only the net effect. This interpretation, therefore, hinges on defining *dependency* or *interaction* based on the existence and intensity of these information flow pathways, rather than strictly on their final, combined impact on a node's prediction.

## I  ADDITIONAL DEFINITIONS IN SECTION 6

**Definition I.1** (Standard lattice in $\mathbb{R}^D$). The *standard lattice* in $\mathbb{R}^D$, denoted by $\mathbb{Z}^D$, is the set of all integer-coordinate points in $\mathbb{R}^D$: $\mathbb{Z}^D = \{(z_1, z_2, \ldots, z_D) \mid z_i \in \mathbb{Z} \text{ for all } i = 1, \ldots, D\}$. Equivalently, $\mathbb{Z}^D$ consists of all points that can be written as integer linear combinations of the standard basis vectors: $\mathbb{Z}^D = \left\{\sum_{i=1}^{D} z_i \boldsymbol{e}_i \mid z_i \in \mathbb{Z}\right\}$, where $\{\boldsymbol{e}_1, \boldsymbol{e}_2, \ldots, \boldsymbol{e}_D\}$ is the standard basis for $\mathbb{R}^D$, meaning each $\boldsymbol{e}_i$ is a unit vector with a 1 in the $i$-th position and 0 elsewhere. This lattice forms a grid-like structure in $\mathbb{R}^D$ with each point having exactly $2D$ one-hop (adjacent lattice) neighbors. For a planar graph $D = 2$, hence, each node has a total of 4 neighbors.

**Definition I.2** ($h$-hop shells). Let $\mathcal{G} = (V, E)$ be a graph with shortest-path distance $\rho : V \times V \to \mathbb{N}$. The *$h$-hop shell* (or *$h$-hop neighborhood*) of a node $v \in V$ is defined as $\mathcal{N}_h(v) = \{u \in V : \rho(v, u) = h\}$. That is, $\mathcal{N}_h(v)$ consists of all nodes that are exactly $h$ hops away from $v$.

**Definition I.3** (Quasi-isometric graphs). Let $\mathcal{G}_1 = (V_1, E_1)$ and $\mathcal{G}_2 = (V_2, E_2)$ be two graphs equipped with shortest-path distance functions $\rho_1 : V_1 \times V_1 \to \mathbb{R}_{\geq 0}$ and $\rho_2 : V_2 \times V_2 \to \mathbb{R}_{\geq 0}$, respectively (in our case distances are in $\mathbb{N}$). We say that $\mathcal{G}_1$ and $\mathcal{G}_2$ are *quasi-isometric* if there exist constants $\lambda \geq 1$, $\mathfrak{C} \geq 0$, and $\mathfrak{D} \geq 0$, and a function $f : V_1 \to V_2$ such that for all $u, v \in V_1$: $\frac{1}{\lambda}\rho_1(u, v) - \mathfrak{C} \leq \rho_2(f(u), f(v)) \leq \lambda\rho_1(u, v) + \mathfrak{C}$, Every node in $V_2$ is within distance $\mathfrak{D}$ of some $f(u)$, i.e., $\forall v \in V_2, \exists u \in V_1$ such that $\rho_2(v, f(u)) \leq \mathfrak{D}$.

## J  PROOFS FOR THEORETICAL RESULTS IN SECTION 6

*Proof of Lemma 6.1.* Since $\mathcal{G}$ is grid-like in $D$ dimensions, its structure mimics that of $\mathbb{Z}^D$. In $\mathbb{R}^D$, the volume of a ball of radius $h$ scales as $h^D$ (recall the volume of a sphere in $\mathbb{R}^3$ is $\frac{4}{3}\pi h^3$). Let

$$B_h(v) = \{u \in V : \rho(v, u) \leq h\}, \tag{47}$$

denote the ball of radius $h$ centered at $v$. Then for large $h$ (asymptotic bound),

$$|B_h(v)| = \Theta(h^D). \tag{48}$$

Since the $h$-hop neighborhood is the set difference

$$\mathcal{N}_h(v) = B_h(v) \setminus B_{h-1}(v). \tag{49}$$

This effectively means that the size of $\mathcal{N}_h(v)$ is approximately the difference between the volume of two consecutive balls:

$$|\mathcal{N}_h(v)| = |B_h(v)| - |B_{h-1}(v)|. \tag{50}$$

A standard asymptotic argument implies

$$|\mathcal{N}_h(v)| = \Theta(h^{D-1}), \tag{51}$$

since we can approximate the aforementioned difference via:

$$|B_h(v)| - |B_{h-1}(v)| = \mathfrak{C}h^D - \mathfrak{C}(h-1)^D \approx \mathfrak{C}h^D - \mathfrak{C}(h^D - Dh^{D-1}) = \mathfrak{C}Dh^{D-1}, \tag{52}$$

using the binomial expansion and assuming large $h$ (we are concerned with long-range interactions), where the leading order term dominates. Intuitively, this corresponds to the *surface growth* of the ball, which in $\mathbb{R}^D$ scales as $h^{D-1}$ (the area of a sphere in $\mathbb{R}^3$ is $4\pi h^2$). Thus, there exist constants $\mathfrak{C}_1, \mathfrak{C}_2 > 0$ and an integer $h_0$ such that for all $h \geq h_0$,

$$\mathfrak{C}_1 \, h^{D-1} \leq |\mathcal{N}_h(v)| \leq \mathfrak{C}_2 \, h^{D-1}. \tag{53}$$

$\square$

*Proof of Theorem 6.2.* Assume that within $\mathcal{N}_h(v)$ there is a unique node $u^*$ with a strong influence $I(v, u^*) = I^* > 0$ and that for all other nodes $u \in \mathcal{N}_h(v) \setminus \{u^*\}$, the influence $I(v, u)$ is negligible ($\Delta \approx 0$). Then:

The *total* (or *sum*) influence is

$$I_{\text{sum}}(v, h) = \sum_{u \in \mathcal{N}_h(v)} I(v, u) \geq I(v, u^*) = I^*. \tag{54}$$

The *mean* influence is given by

$$I_{\text{mean}}(v, h) = \frac{1}{|\mathcal{N}_h(v)|} \sum_{u \in \mathcal{N}_h(v)} I(v, u) = \frac{1}{|\mathcal{N}_h(v)|}(I^* + \sum_{u \in \mathcal{N}_h(v)|u^*} I(v, u)) = \frac{I^* + \Delta}{|\mathcal{N}_h(v)|}. \tag{55}$$

Using the lower bound on $|\mathcal{N}_h(v)|$,

$$I_{\text{mean}}(v, h) \leq \frac{I^* + \Delta}{\mathfrak{C}_1 \, h^{D-1}}. \tag{56}$$

Since $h^{D-1} \to \infty$ as $h \to \infty$ for $D \geq 2$, it follows that

$$I_{\text{mean}}(v, h) \to 0, \tag{57}$$

while $I_{\text{sum}}(v, h) \geq I^*$ remains non-vanishing. $\square$

*Proof of Corollary 6.3.* For a planar graph that is grid-like (for example, a two-dimensional lattice), set $D = 2$. Then, $|\mathcal{N}_h(v)| = \Theta(h^{2-1}) = \Theta(h)$. Repeating the same argument as above: the total influence satisfies $I_{\text{sum}}(v, h) \geq I^*$. The mean influence is bounded by $I_{\text{mean}}(v, h) \leq \frac{I^*}{|\mathcal{N}_h(v)|} \leq \frac{I^*}{\mathfrak{C}_1 h}$. Hence, as $h \to \infty$, $I_{\text{mean}}(v, h) \to 0$, which demonstrates that for a planar graph the dilution of the mean aggregated influence occurs at a rate proportional to $1/h$. $\square$

*Proof of Corollary 6.4.* Since $\mathcal{G}$ is grid-like in $D$ dimensions, the size of the aggregated $h$-hop neighborhood (the ball) grows as $|B_h(v)| = \Theta(h^D)$. Assuming that only one node $u^*$ in $B_h(v)$ has a significant influence $I^*$ while the influence of all other nodes is negligible, the total (or sum) influence satisfies $I_{\text{sum}}^B(v, h) \geq I^*$. Thus, the mean influence over $B_h(v)$ is $I_{\text{mean}}^B(v, h) \leq \frac{I^*}{\Theta(h^D)}$. That is, there exists a constant $\mathfrak{C}' > 0$ such that $I_{\text{mean}}^B(v, h) \leq \frac{I^*}{\mathfrak{C}' h^D}$. Since $h^D \to \infty$ as $h \to \infty$ for $D \geq 1$, it follows that $I_{\text{mean}}^B(v, h) \to 0$. $\square$

*Proof of Corollary 6.5.* For any node $v \in V$, by the distinguished node assumption (reused from Theorem 6.2) there is at least one $u^* \in \mathcal{N}_h(v)$ with

$$I(v, u^*) \geq I^*. \tag{58}$$

Since the total influence is a sum over nonnegative contributions, it follows that

$$T_h(v) = \sum_{u \in \mathcal{N}_h(v)} I(v, u) \geq I(v, u^*) \geq I^*. \tag{59}$$

Averaging over all nodes in $V$,

$$\overline{T}_h = \frac{1}{|V|} \sum_{v \in V} T_h(v) \geq \frac{1}{|V|} \sum_{v \in V} I^* = I^*. \tag{60}$$

This bound is independent of the size $|\mathcal{N}_h(v)|$ of the $h$-hop neighborhood and hence remains valid even as $|\mathcal{N}_h(v)|$ (or the overall number of nodes) tends to infinity.

$$\overline{T}_h = \frac{1}{|V|} \sum_{v \in V} T_h(v) \geq I^*, \quad \forall h. \tag{61}$$

$\square$

