# OpenReview forum: "Towards Quantifying Long-Range Interactions in Graph Machine Learning: a Large Graph Dataset and a Measurement"
_ICLR.cc/2026/Conference — ICLR 2026 Poster_

### Official Review · Reviewer_NW7j · 2025-10-22

**Soundness:** 4
**Presentation:** 4
**Contribution:** 4
**Rating:** 8
**Confidence:** 4

**Summary:**

The paper suggests a new graph dataset that is clearly based long-range dependencies. The graphs are based on city road networks of four major cities with the target to compute the furthest distance one can travel passing 16 junctions. In addition, they provide a measurement that computes the influence of far-away nodes, highlighting that indeed the constructed graphs contain long-range dependencies.

**Strengths:**

The paper addresses an important open problem in graph learning as currently good benchmarks for long-range dependencies are missing even though we know (or rather assume) that long-range dependencies exist in many real-world tasks. The construction of the dataset is very clear and the provided metric improves upon the main existing alternative (Bamberger et al 2025) in terms of speed and possibly accuracy. The theoretical justification is non-trivial and makes a sound impression. The paper is well-written and easy to follow.

**Weaknesses:**

Not exactly strong weaknesses, but rather points that I would have liked:
- a more detailed comparison to the metric by Bamberger et al which I did expect to be mentioned in the introduction as well (e.g. in 107 I did expect that reference)
- a more concrete statement in 125ff that the appendix contains the exact list of features that made it into the dataset. The description in the main paper is a little to vague here for my personal taste.
- the conclusion states that LRGB's claim for long-range is solely based on the performance gap while the paper also talks about larger (even though not large) graphs and larger diameter.

**Questions:**

see weaknesses.

---

> ### Author Response · Authors · 2025-11-20
> **Author's Response to Reviewer NW7j**
>
> We thank the reviewer for the positive evaluation of our work! Please see our detailed response below.
>
> ---
>
> > A more detailed comparison to the metric by Bamberger et al, which I did expect to be mentioned in the introduction as well (e.g. in 107 I did expect that reference)
>
> ***Re:*** Thank you for raising this point. The metric proposed by the concurrent work of Bamberger et al. (2025) is indeed related to the metric we propose, in the sense that both rely on the calculation of influence scores via the Jacobian-based analysis. In particular, the definition of R defined in Eq.(3) of our paper uses shortest path distance, which corresponds to a specific instance of the measure in Bamberger et al. (2025).
>
> The main difference between the two research is that the former is mainly interested in an aggregated measure of the range of a model when solving a task, which answers “how long-range does the model reach on average?”. On the other hand, the latter has a particular focus on using the per-hop influence $T_h(v)$ as a diagnostic tool to analyse dependency decay along the hops, which leads to key analyses in Fig.4.
>
> In addition, Bamberger et al. (2025) focus on a theoretical characterisation of the range measure, where our main focus is on novel large-scale long-range graph datasets. We will discuss these relations in more detail in the revised version and also properly mention Bamberger et al. (2025) in the Introduction.
>
> ---
>
> > A more concrete statement in 125ff that the appendix contains the exact list of features that made it into the dataset. The description in the main paper is a little too vague here for my personal taste.
>
> ***Re:*** We thank the reviewer for this suggestion. Due to the page limit, we presented a summary of the features and briefly explained our feature engineering approach in the main text. We will add a more concrete statement in the revised version and make sure this information is clear in the main text.
>
> ---
>
> > The conclusion states that LRGB's claim for long-range is solely based on the performance gap, while the paper also talks about larger (even though not large) graphs and larger diameter.
>
> ***Re:*** We agree with the reviewer that the LRGB paper also mentioned that one consideration behind their dataset design is the diameter of the graph, even though the graphs they considered are on average of much smaller sizes (hence smaller diameters). We will add this to the relevant discussion and properly credit the LRGB datasets in the revised version.
>
> ---
>
> We'd like to thank the reviewer again for these suggestions and positive reviews of our work.
>
> **Reference**
>
> [1] On Measuring Long-Range Interactions in Graph Neural Networks. ICML 2025

---

### Official Review · Reviewer_e2fb · 2025-10-31

**Soundness:** 3
**Presentation:** 3
**Contribution:** 3
**Rating:** 6
**Confidence:** 3

**Summary:**

This work proposes a real-world, attributed, transductive benchmark consisting of four city networks, where the task is to classify each junction’s "urban accessibility", defined via a local eccentricity measure with radius 16. This, they argue, creates non-trivial long-range task dependencies. They also explore other radii values: smaller ones make the task too easy, while larger ones suffer from uninformativeness of the graph structure. They compare several network metrics with existing graph datasets and show that their city networks have larger diameters but are sparser and more grid-like; this mitigates over-smoothing and makes them better suited for benchmarking long ranges. They benchmark both GNNs and graph transformers and find that, on their dataset, performance consistently improves with depth, whereas on common benchmarks it typically plateaus or drops. Finally, they introduce a measure of long-range dependency that examines the layer-wise and cumulative influence of distant nodes on predictions. On their dataset, this influence is stronger and decays much more slowly than on the others.

**Strengths:**

- The paper introduces a transductive benchmark built from real road networks, and moves beyond the small citation-like graphs that dominate current GNN evaluations of long range tasks.
- On their benchmark, all models improve with depth, which is not the case on most standard datasets, and is thus a good indicator that the task needs long-range interactions. They furthermore support it with theory on over-smoothing, showing why this is possible on this kind of structure.
- Road networks are an important application area of graph learning, so a benchmark based on them is very welcome.

**Weaknesses:**

1. I am not fully convinced by the benefits of the proposed long-range influence metric. It does not seem strictly model-agnostic, since the influence values still depend on the trained parameters and even change across models (cf. Table 2). It mostly answers "did this model use long-range information?" rather than "is this dataset inherently long-range?"
2. I think the limitation of this metric on dense graphs is too important to be buried in the appendix; it should be brought into the main text and discussed more lengthily and explicitly.
3. While it is a good point to test graph transformers on large graphs for scalability, I am not sure this setting is the most revealing for getting insights. I.e. failure cases on smaller graphs are often more informative because there are fewer confounding factors. This benchmark awkwardly sits between being realistic and being revealing.
4. The defined task is not necessarily informing about the accessibility of a city area. For instance, in practice longer and faster highways can make areas more accessible and not less. The task is still a synthetic one defined on top of a real graph structure.
5. Fixing the ground-truth radius at 16 and mostly benchmarking models up to that depth is not entirely fair. The other tasks can make models suffer from over-smoothing from having more layers than required. Apart from a single point in Fig. 8, I would like to see more results with deeper models than the target radius.

**Questions:**

1. It would help to see how the influence measurement behaves on synthetic long-range tasks (1). In a setting without real-world noise and with fully controlled dependencies, does the influence become more stable across models, or do we still observe the same variability?
2. The paper states that "graphs with long-range dependencies are expected to have higher proportional influence between more distant nodes", but this assumes a monotone growth of influence with distance. In practice a task could depend on radius 0 and on exactly radius R, and your global metric would average it out. Can this happen for your task, and could models overfit to intermediate radii in a way that even distorts the per-hop measurement?
3. How does the theoretical argument about sparsity and slower over-smoothing relate to phenomena like the Braess paradox, where removing edges can increase leading eigenvalues, and to empirical findings that sparsity can reduce measured over-smoothing even when the leading eigenvalue grows (2)?

(1) GLoRa: A Benchmark to Evaluate the Ability to Learn Long-Range Dependencies in Graphs. Dongzhuoran Zhou et al., ICLR 2025.

(2) Spectral Graph Pruning Against Over-Squashing and Over-Smoothing. Adarsh Jamadandi et al., NeurIPS 2024.

---

> ### Author Response · Authors · 2025-11-20
> **Author's Rebuttal to Reviewer e2fb (1/4)**
>
> We thank the reviewer for the positive evaluation and the detailed feedback that helped us improve our work. Please find our detailed response below.
>
> ---
>
> > The proposed long-range influence metric does not seem strictly model-agnostic, since the influence values still depend on the trained parameters and even change across models. It mostly answers "did this model use long-range information?" rather than "is this dataset inherently long-range?"
>
> ***Re:*** Thank you for pointing this out: we agree that the phrasing “model-agnostic” may be misleading. Our intention was not to claim that the influence metric produces identical values across different architectures. Rather, “model-agnostic” was meant in the standard sense used in works such as MAML [1]: the metric can be applied to any differentiable GNN or Graph Transformer, without requiring architectural modifications or assumptions about the model class.
>
> As the reviewer correctly notes, the influence score answers the question “did this trained model use long-range information?”, which we use as a data-driven proxy for the underlying long-range nature of the task or dataset. Since the true target function is unknown, any attempt to measure intrinsic long-range dependence must rely on models trained to approximate it. In this setting, evaluating multiple diverse architectures (some of which are explicitly designed to capture long-range signals) allows us to compare their long-range usage relative to one another. We view this comparative perspective as a first step toward characterizing whether a dataset induces long-range dependencies.
>
> We agree that a model incapable of capturing long-range interactions will naturally obtain a low influence score even if the true function is long-range. We acknowledge this as a limitation. Ultimately, obtaining an absolute measure of “true” long-range dependency may require access to the ground-truth function itself, which is precisely the latent object we hope the model to be able to learn. We therefore believe our approach is a practical intermediate solution.
>
> To avoid confusion, we are happy to replace “model-agnostic” with a more accurate term, such as architecture-general, architecture-independent to apply, or model-compatible, and can clarify this via a footnote.
>
> ---
>
> > While it is a good point to test graph transformers on large graphs for scalability, I am not sure this setting is the most revealing for getting insights; i.e., failure cases on smaller graphs are often more informative because there are fewer confounding factors.
>
> ***Re:*** We agree that failure cases on small graphs can be informative, and we view our large-graph evaluation as complementary rather than a replacement. Our results show that even graph transformers specifically designed for scalability still struggle to capture long-range interactions in large graphs: a limitation that often does not appear on smaller benchmarks. This is largely because positional encodings that are tractable on small graphs become computationally prohibitive at our scale. Without such encodings, the attention mechanism cannot effectively “count” hops and must rely solely on the available node features.
>
> In our dataset, labels depend on 16-hop structural information. While geographic coordinates (longitude/latitude) can provide some weak global cues, they are insufficient for identifying 16-hop relationships because graph distance and Euclidean distance diverge significantly (see Figure 6). In some parts of the network, 16 hops correspond to a very small Euclidean radius due to dense road structure, while in others the same hop distance spans a much larger geographic area. Because the models lack positional encodings, graph transformers cannot exploit this structural information any better than GNNs, which is reflected in their similarly limited performance.
>
> These insights are, in our view, unique to large-graph settings and reveal model weaknesses that small-graph benchmarks cannot expose. At the same time, to address the reviewer’s concern (and as part of our response to reviewer ckYP), we have added experiments on small heterophilic graphs. Our paper also includes evaluations on Cora, PascalVOC-SP, and Amazon-Ratings, all of which are small-scale benchmarks, thereby providing complementary insights across graph sizes.

---

> ### Author Response · Authors · 2025-11-20
> **Author's Rebuttal to Reviewer e2fb (2/4)**
>
> > I think the limitation of this metric on dense graphs is too important to be buried in the appendix; it should be brought into the main text and discussed more lengthily and explicitly.
>
> ***Re:*** We thank the reviewer for the suggestion of a more detailed discussion on the limitation.
>
> We state in our paper that measuring long-range dependency on large, dense graphs often requires expensive computation. For example, ogbn-arxiv has 169k nodes with an average degree of 14 and a diameter of 25, where a 16-hop neighborhood covers over 90% of the network on average. In this case, the computational cost $O(NN_H)$ approaches $O(N^2)$, which is often impractical with a complex model.
>
> In our paper, we moved the explanation to the appendix due to limited space in line with the ICLR guidelines. We will add a remark on this at the end of Section 4 in the revised version and discuss this issue more explicitly. Meanwhile, we also want to point out that this is a common challenge for all Jacobian-based analysis in the literature [5, 6], and it is an open question to design an efficient algorithm for approximating such long-range influence in this case.
>
> ---
>
> > The defined task is not necessarily informing about the accessibility of a city area. For instance, in practice longer and faster highways can make areas more accessible and not less.
>
> ***Re:*** We thank the reviewer for pointing this out. We agree with the reviewer that in real-world transportation networks, accessibility is more than just hop‐count distance: factors like road speed, capacity, and highway connectivity strongly influence how “accessible” one area is from another. In our current dataset, we simplify by assuming a constant travel speed. This simplification gives us a clean and controllable setup to study long-range topological interactions, but it does mean we abstract away some real-world transport dynamics.
>
> That said, this suggestion is well taken, and we believe our framework can be extended in a natural way by using road_length/speed_limit to approximate the travel time (or use other travel-time proxies) in defining the edge weights in Eq. (1). Due to the limited time of rebuttal, we present some preliminary results on Paris below under this labeling strategy. We can see that the trend of performance when increasing the model depth is consistent with our original labelling approach.
>
> | Paris | 2          | 4          | 8          | 16         |
> | ------------- | ---------- | ---------- | ---------- | ---------- |
> | **GCN**       | 26.9 ± 0.3 | 32.4 ± 0.4 | 42.6 ± 0.3 | 49.8 ± 0.4 |
> | **GraphSAGE** | 25.9 ± 0.4 | 36.6 ± 0.3 | 47.3 ± 0.4 | 51.6 ± 0.5 |
>
>
> ---
>
> > Fixing the ground-truth radius at 16 and mostly benchmarking models up to that depth is not entirely fair. The other tasks can make models suffer from over-smoothing from having more layers than required. Apart from a single point in Fig. 8, I would like to see more results with deeper models than the target radius.
>
> ***Re:***
> One of the advantages of adopting a fixed range is that we can confirm and control the ground-truth range, so that we can avoid our long-range task being "the longer the communication path the better." With that being said, we agree with the reviewer that benchmarking GNNs with depths deeper than the ground-truth signal range (16th hop) will help us better understand the model’s behaviour. As suggested, we further test GCN and GraphSAGE with depth=[32, 48, 64] on Paris and Shanghai under the same experimental settings in the paper. The results are summarized in the tables below. As expected, we can observe that both GNNs (with residual connections and batch norm) start to suffer from oversmoothing when model depth exceeds the ground-truth range.
>
> | Paris | 16         | 32         | 48         | 64         |
> | ------------- | ---------- | ---------- | ---------- | ---------- |
> | **GCN**       | 53.2 ± 0.3 | 50.1 ± 0.4 | 48.7 ± 0.4 | 47.6 ± 0.3 |
> | **GraphSAGE** | 54.6 ± 0.2 | 48.9 ± 0.3 | 47.1 ± 0.3 | 44.3 ± 0.5 |
>
> | Shanghai | 16         | 32         | 48         | 64         |
> | ------------- | ---------- | ---------- | ---------- | ---------- |
> | **GCN**       | 62.1 ± 0.2 | 57.7 ± 0.3 | 53.8 ± 0.3 | 52.4 ± 0.4 |
> | **GraphSAGE** | 68.3 ± 0.5 | 57.1 ± 0.4 | 51.6 ± 0.4 | 49.3 ± 0.3 |

---

> ### Author Response · Authors · 2025-11-20
> **Author's Rebuttal to Reviewer e2fb (3/4)**
>
> > It would help to see how the influence measurement behaves on synthetic long-range tasks [2]. In a setting without real-world noise and with fully controlled dependencies, does the influence become more stable across models, or do we still observe the same variability?
>
> ***Re:*** We thank the reviewer for the suggestion on synthetic tasks. Unfortunately, when we try to implement the algorithm in GLoRa [2], we are unable to access their GitHub repo (404 error as of November 20, 2025) using the link provided in their paper (https://github.com/DongzhuoranZhou/GLoRa). Alternatively, we try our measurement on the RingTransfer [3] experiment, which is used for testing long-range dependency in GNNs under inductive settings.
>
> **RingTransfer.** Each graph in the RingTransfer dataset is a ring of $k$ nodes with only two nodes marked: the source node and the target node, which are placed at opposite sides of the ring (i.e., at a distance of the diameter $k/2$). All nodes on the ring will have a constant feature vector except for the source node, which has a one-hot encoding of its label. The task is to train a model such that the target node’s representation predicts the source’s label, which requires the model to propagate long-range information from the source node to the target node.
>
> **Our setups.** We followed the same setups in [3] and adopted k=16 (i.e., a signal range of 8 hops) with a model depth of 8 in our setting, as we observed a similar phenomenon in [3] that GNNs start to deteriorate after this point. In particular, we evaluate GCN, GraphSAGE, and Exphormer with training/validation/test splits of 5k/1k/1k graphs, where all models can achieve 100% accuracy on the testing set. We then apply our measurement on the target node only for each graph, and then report the average over the testing set.
>
> **Results and discussion.** The results for R and per-hop influence are summarized below. As expected, we can observe a high influence at hop 8 for GCN and GraphSAGE; while for Exphormer, as it adopts global virtual node and expander graph operations in its attention mechanism, the source node can be effectively reached within a single hop. Therefore, we observe a higher influence on the first few hops compared to more distant hops since the underlying graph has been modified.
>
> While we do observe different influence patterns due to different model designs, we'd also like to point out that this is because RingTransfer is too simplistic and relies solely on long-range interactions, such that simple operations (e.g., rewiring, virtual node, etc.) will convert it into a short-range task.
>
> | Model / Hop | 0 | 1    | 2    | 3    | 4    | 5    | 6    | 7    | 8     |
> | ------------- | - | ---- | ---- | ---- | ---- | ---- | ---- | ---- | ----- |
> | **GCN**       | 1 | 2.12 | 2.05 | 2.05 | 2.61 | 3.53 | 9.43 | 8.14 | 15.20 |
> | **GraphSAGE** | 1 | 1.27 | 2.02 | 0.84 | 1.42 | 0.97 | 4.80 | 3.12 | 5.48  |
> | **Exphormer** | 1 | 2.18 | 1.79 | 1.68 | 1.08 | 1.08 | 1.19 | 0.79 | 0.59  |
>
> |       | GCN  | GraphSAGE | Exphormer |
> | ----- | ---- | --------- | --------- |
> | **R** | 5.27 | 5.21      | 3.34      |
>
> ---
>
> > The paper states that "graphs with long-range dependencies are expected to have higher proportional influence between more distant nodes", but this assumes a monotone growth of influence with distance. In practice a task could depend on radius 0 and on exactly radius R, and your global metric would average it out. Can this happen for your task, and could models overfit to intermediate radii in a way that even distorts the per-hop measurement?
>
> ***Re:*** We thank the reviewer for this insightful point. In Section 4, our intention in the statement is to say that R will be larger in long-range tasks compared to short-range tasks, since the influence from distant nodes to the target node will be proportionally higher if the signal is long-range. While we do not assume a monotonic growth of the influence with distance, in our road networks, intermediate hops generally provide partial information about the labels, so the influence will roughly grow monotonically with distance up to the true range. We will make sure this statement is clear in the revised version.
>
> In addition, while it is theoretically possible for a task to depend sharply on nodes at exactly a certain radius, GNNs and graph transformers that incorporate a message-passing module have an inherent diffusion-like inductive bias: messages propagate and aggregate across neighborhoods, naturally smoothing the learned function over hops. As a result, even if the true dependence is “spiky”, the per-hop influence scores measured from the learned models are unlikely to show sharp discontinuities, and the influence of intermediate nodes can act as a meaningful proxy.

---

> ### Author Response · Authors · 2025-11-20
> **Author's Rebuttal to Reviewer e2fb (4/4)**
>
> > How does the theoretical argument about sparsity and slower over-smoothing relate to phenomena like the Braess paradox, where removing edges can increase leading eigenvalues, and to empirical findings that sparsity can reduce measured over-smoothing even when the leading eigenvalue grows [4]?
>
> ***Re:*** Thank you for this thoughtful comment. There are two relations to be clarified here: 1) the sparsity of the graph and the spectral gap (or algebraic connectivity, or what is referred to as “leading eigenvalue” in [4]); 2) the spectral gap and the tendency of over-smoothing. We discuss both in turn as follows.
>
> Regarding 1): our main theoretical result in Section 5 (Theorem 5.2) suggests that the second largest eigenvalue of the normalised augmented adjacency matrix has a lower bound that is increasing in graph diameter and decreasing in maximum node degree, hence the spectral gap tends to be smaller for graphs with larger diameter and smaller maximum degree. However, we would like to clarify that this tendency may not be exact as the bound is not necessarily tight, and furthermore, a smaller maximum degree does not necessarily imply a sparser graph (although the two can be generally correlated). Therefore, our result is not in contradiction with the Braess paradox, which is an interesting observation by itself. Nevertheless, we acknowledge this is a good point and will add a discussion in the revised version (in particular, we will be more specific when we mention “sparse graphs”).
>
> Regarding 2): our analysis of the rate of over-smoothing is purely based on the spectral analysis that the speed of convergence to a stationary point upon repeated application of a matrix operator depends on the spectral gap: a smaller gap generally slows down convergence in this precise sense. It is worth noting that this analysis is **task-agnostic**, i.e., it does not take into account node classification as a task, and furthermore distribution of node labels. The analysis of over-smoothing in [4], according to our understanding, is, however, **task-dependent**, as it specifically highlights situations when pruning edges can mitigate over-smoothing of task-beneficial signals by disconnecting nodes with different labels. Therefore, while both are meaningful, the two analyses look at over-smoothing from slightly different perspectives. Nevertheless, we again acknowledge this as a good point and will add a discussion in the revised version.
>
> ---
> \
> We would like to thank the reviewer again for providing these constructive comments that helped us improve our work. We believe we have now addressed all the concerns raised in the reviews, and we hope our replies will affirm the reviewer’s positive evaluation of our work.
>
> \
> **Reference**
>
> [1] Model-Agnostic Meta-Learning for Fast Adaptation of Deep Networks. ICML 2017.
>
> [2] GLoRa: A Benchmark to Evaluate the Ability to Learn Long-Range Dependencies in Graphs. ICLR 2025.
>
> [3] Weisfeiler and Lehman Go Cellular: CW Networks. NeurIPS 2021.
>
> [4] Spectral Graph Pruning Against Over-Squashing and Over-Smoothing. NeurIPS 2024.
>
> [5] Representation Learning on Graphs with Jumping Knowledge Networks. ICML 2018.
>
> [6] Influence-Based Mini-Batching for Graph Neural Networks. LoG 2022.

---

> ### Comment · Reviewer_e2fb · 2025-11-26
>
> I thank the authors for their detailed responses. I would like to see some more revisions implemented in the manuscript, as I believe there are still important changes to the storyline and clarity needed to fully address my previously raised concerns.
>
> W1: I consider this nuance in the definition and use of the metric to be very important throughout the paper, and I would like the descriptions to be updated accordingly (not only the term itself). As I understand it, the paper already uses a model-dependent quantity that is quite effective at answering whether the model uses long-range information, namely the accuracy of the model as a function of layer depth. But currently, the argument seems somewhat circular: accuracy is used to justify the measurement, and the measurement to justify accuracy. For this reason, I remain unconvinced that the proposed measurement is as central a contribution as the work suggests.
>
> W3: If the main purpose of the benchmark is to demonstrate that graph transformers fail to capture long-range interactions on larger graphs (because computing full encodings is not feasible at scale), this should be made more central in the narrative and in the stated motivation for the benchmark. However, I am not sure this observation is very surprising, given that (1) already criticizes the performance of GTs on long-range interactions. I understand that their graph structures and tasks are quite different, but the resulting insight seems similar. It might help to narrow down the contribution by clarifying: What would be required for a model to perform better on this benchmark?
>
> W4: Thank you for providing results with a different edge weight definition. This addresses the specific example I mentioned, but not the underlying concern: the benchmark remains predominantly synthetic, since these metric are still computed from the graph rather than directly measured from the real world.
>
> W5: Thank you for the results at higher depths. I am not sure I would describe the behaviour as over-smoothing if performance happens to peak at the known ground-truth depth in all cases. At these high depths, one could also interpret this as overfitting on information coming from higher layers. How does the proposed measurement behave in this regime? Does it incorrectly attribute useful signal to information coming from higher depths? If so, this would represent another limitation of the measurement that should be discussed.
>
> Q1: Thank you for trying another synthetic example; that is sufficient for me. But again the results are only shown up to the ground-truth depth but not beyond it.
>
> Q2: To me, this also represents a limitation of the measurement that should be explicitly discussed.
>
> Q3: Thank you for the added discussion; I agree with the conclusions.

---

> > ### Author Response · Authors · 2025-12-01
> > **Response to Reviewer e2fb (3/3)**
> >
> > > Q1: Thank you for trying another synthetic example; that is sufficient for me. But again the results are only shown up to the ground-truth depth but not beyond it.
> >
> > ***Re:*** We thank the reviewer for noticing this detail. In the RingTransfer experiment, both signal range and graph diameter are $N/2=8$ on a ring of $N=16$ nodes, which means there will be no gradient w.r.t. nodes beyond the 8th hop, even if the model has a depth deeper than 8. For your reference, we have also tested models with 16 layers and computed their influence up to 8 hops, where we observe very similar results to those from models with 8 layers.
> >
> > ---
> >
> > We thank the reviewer for their response to our rebuttal. We believe our additional clarifications address the remaining questions and concerns, and we hope this strengthens confidence in the reviewer’s positive evaluation of our work.

---

> ### Author Response · Authors · 2025-12-01
> **Response to Reviewer e2fb (1/3)**
>
> We’d like to thank the reviewer for their follow-up comments to our rebuttal. The modifications are now all updated in the revised PDF (highlighted in blue). Please see our response to the questions below.
>
> ---
>
> > W1: I consider this nuance in the definition and use of the metric to be very important throughout the paper, and I would like the descriptions to be updated accordingly (not only the term itself).
>
> ***Re:*** We thank the reviewer for the suggestion, and we have now changed the term to “a generic measurement” with an updated description in the introduction.
>
> ---
>
> > As I understand it, the paper already uses a model-dependent quantity that is quite effective at answering whether the model uses long-range information, namely the accuracy of the model as a function of layer depth. But currently, the argument seems somewhat circular: accuracy is used to justify the measurement, and the measurement to justify accuracy.
>
> ***Re:*** One of the main objectives of our paper is to introduce a benchmark that requires long-range information over large graphs, which is built into the learning task by design. As a validation, we show that the performance of different models all have a consistent increasing trend on our city networks when the models’ depth increases (up to the ground-truth range), which is not typically observed in existing graph benchmarks. This answers the question “whether the model uses long-range information”, but it is not sufficient to address the question “How long-range does the model reach?”
>
> To answer the second question, we propose a global measurement $R$ to quantify the average range that a model looks at when making predictions, and use a per-hop influence metric $T_h$ to further explain the model’s behavior. These two diagnostic tools enable us to quantitatively compare the long-range patterns across (1) different models on the same dataset, and (2) different datasets under the same model, which is **NOT** achievable by only looking at the performance trend vs model depth.
>
> For example, while all models have similar upward performance trends on our city networks, we found that $R$s for GPS and Exphormer are generally larger than GCN and GraphSAGE due to the difference in architectures. Moreover, although heterophilic graphs have been used for testing models’ long-range capability in the literature, recent works [1] start to question the relationship between these two concepts, and we are the first to empirically show that heterophily does not necessarily imply long-range dependency. In particular, we found that $R$s on Amazon-Ratings (Table 2 in Section 4), Roman-Empire, Texas, Wisconsin, and Cornell (Table 15 in Appendix C.6, newly added during rebuttal) are even smaller than Cora for most of the baselines we tested.
>
> In summary, we did not use the accuracy to justify our measurement. But instead, we use the measurement as a complementary tool to help us better understand the underlying behavior of different models across different datasets, which, by itself, is a meaningful contribution to the literature for analyzing long-range dependency.
>
> [1] Oversmoothing, “Oversquashing”, Heterophily, Long-Range, and more: Demystifying Common Beliefs in Graph Machine Learning. arXiv:2505.15547.
>
> ---
>
> > W3: If the main purpose of the benchmark is to demonstrate that graph transformers fail to capture long-range interactions on larger graphs (because computing full encodings is not feasible at scale), this should be made more central in the narrative and in the stated motivation for the benchmark.
>
> ***Re:*** We thank the reviewer for this suggestion, and we have made this information clearer in the introduction when stating our motivation. At the same time, we also want to clarify that we did not state that GTs “fail” to capture long-range interactions on large graphs, but rather are less effective in modelling long-range information compared to tasks on small graphs due to the impractical computation of positional encodings. As a result, unlike LRGB, we do not observe a clear difference in performance between GTs and GNNs on our city networks.

---

> ### Author Response · Authors · 2025-12-01
> **Response to Reviewer e2fb (2/3)**
>
> > However, I am not sure this observation is very surprising, given that (1) already criticizes the performance of GTs on long-range interactions. I understand that their graph structures and tasks are quite different, but the resulting insight seems similar. It might help to narrow down the contribution by clarifying: What would be required for a model to perform better on this benchmark?
>
> ***Re:*** We agree with the reviewer that our work shares some similarity with GLoRa [1], since both works are about evaluating long-range dependency. For example, on both datasets, the models need to propagate long-range information to make accurate predictions, which requires deeper depths compared to tasks on short-ranged datasets.
>
> However, our city networks feature a larger graph size at order $N(10^6)$ and a large diameter from 100 to 400 compared to existing long-range benchmarks such as LRGB and GLoRa, where the graph size is either at $N(10)$ or $N(10^2)$ and the diameter is at $N(10)$. This unique graph structure requires the architecture to be scalable on large graphs while effectively capturing long-range information, which provides a new challenge to the literature.
>
> In terms of "what would be required for a model to perform better on this benchmark", we believe importantly the model must be able to both take into account both node features and graph structure, and capture long-range dependency by making distant nodes communicate with each other. The challenges with GTs on our dataset is partly due to the fact that positional encodings are expensive to compute (which makes GTs less structure-aware).
>
> ---
>
> > W4: Thank you for providing results with a different edge weight definition. This addresses the specific example I mentioned, but not the underlying concern: the benchmark remains predominantly synthetic, since these metrics are still computed from the graph rather than directly measured from the real world.
>
> ***Re:*** We agree with the reviewer that our label construction is not directly observed from the real world, while other labelling methods, such as pair-wise travel time measurement using Google Map API queries, will make the prediction task more realistic. We have added a discussion on this limitation in Appendix B.2 in the revised version.
>
> Meanwhile, we also want to reiterate that our goal here is to strike a balance between being fully realistic and being controllable over the signal range, so that we can avoid our long-range task simply being "the longer the communication path the better” during benchmarking and post-analysis.
>
> ---
>
> > W5: Thank you for the results at higher depths; I would like these to be included in the revision. I am not sure I would describe the behaviour as over-smoothing if performance happens to peak at the known ground-truth depth in all cases. At these high depths, one could also interpret this as overfitting on information coming from higher layers. How does the proposed measurement behave in this regime? Does it incorrectly attribute useful signal to information coming from higher depths? If so, this would represent another limitation of the measurement that should be discussed.
>
> > Q2: To me, this also represents a limitation of the measurement that should be explicitly discussed.
>
> ***Re:*** The results for deeper models and their influence analysis are now updated in the revised version in Appendix C.3. Regarding W5 and Q2, we agree with the reviewer that the bias in the model will lead to a limitation in our measurement, which we explain as follows.
>
> When the models’ depth (64 layers) is much deeper than the known signal depth (16 hops), we observe that $R$s are generally larger than those from models with 16 layers in our city networks. Meanwhile, the per-hop influence also suggests that deeper models leverage information from distant hops beyond the ground truth range $k$, which makes sense since the model has no information about $k$, and node features beyond the $k$th hop may contain useful information for prediction. Therefore, the biased approximation of the ground-truth function will inevitably lead to a biased estimation of the underlying signal’s range, even though the measurement remains a faithful description of how the model utilises long-range information. The same logic also applies to the scenario described in Q2 when the true signal is “spiky” on certain hops, but the inductive bias of GNNs smooths out the learned function across all layers.
>
> However, our conclusions regarding long-rangeness across datasets are still valid, since each model is validated across different datasets under the same depth in our analysis, where all models show clear long–range patterns on our city-networks compared to those on the existing benchmarks at depth=16 (Section 4) and depth=64 (Appendix C.3).
>
> We would like to thank the reviewer again for this insightful comment, and we have added a discussion in Appendix E and mentioned this limitation in the main text.

---

### Official Review · Reviewer_ckYP · 2025-11-01

**Soundness:** 4
**Presentation:** 4
**Contribution:** 3
**Rating:** 4
**Confidence:** 2

**Summary:**

This paper introduces City-Networks, a new benchmark dataset designed to evaluate long-range dependency handling in graph learning models. It consists of large real-world road networks (100k–569k nodes) from four global cities and labels nodes by computing local eccentricity over large hop distances (k=16 layers). The task is transductive node classification; models must incorporate information from far-away neighborhoods to succeed.

The authors also propose a model-agnostic Jacobian-based influence measure that quantifies how much distant nodes contribute to predictions. They show deeper GNNs and graph transformers consistently improve on these datasets and provide theoretical justification linking dataset topology to reduced over-smoothing and emphasizing influence dilution in grid-like graphs.

**Strengths:**

1. **Novel Benchmark Contribution**: Introduces a real-world long-range benchmark on large graphs.

2. **Task Design**:
- Long-range target signal (local eccentricity) is explicitly tied to graph distance, not just node features.
- Sensible justification for choosing k=16 to require long-range aggregation.

3. **Good Empirical Study**: Systematic layer-depth experiments show deeper message passing helps.

4. **Theoretical Support**: Spectral argument connecting large diameter & low degree to slower over-smoothing.

**Weaknesses:**

1. **More Clarity Needed on Task Setup**:
- Distribution of quantile labels — class imbalance?
- Exact splits and sampling details

2. **Transductive-only Setting**: Dataset is transductive; how will the ideas generalize to inductive settings?

3. **Label Leakage and Spatial Bias Concerns**: Although the authors argue against pure geographical dependence, node features include latitude & longitude and spatially-derived attributes. It is not fully demonstrated that models can't rely largely on spatial features alone. Stronger evidence is needed to show that spatial coordinates alone cannot solve most of the signal.

**Questions:**

1. Can you please defend and answer the questions or concerns raise in **Weaknesses**?
2. How sensitive is performance to the inclusion of geographic features (lat/long)? Can you report results where positional coordinates are removed?
3. Can models exploit spatial coordinates alone (that is consider separately MLP with coordinates only and GNNs with coordinates masked)?
4. What is the runtime overhead of Jacobian measurement on a single city graph?
5. Can you please provide a Table with accuracy results you presented in Figure 3 for k=16? It will be more easier to compare the results.
6. How would your model perform on heterephilic datasets (Texas, Wisconsin, Cornell, Roman-Empire, etc)

---

> ### Author Response · Authors · 2025-11-20
> **Author's Rebuttal to Reviewer ckYP (1/3)**
>
> We thank the reviewer for the detailed and valuable feedback, which we find very helpful in improving our work! We are also glad to see the reviewer acknowledged the novelty, task design, and soundness of the empirical study in our paper! Please find our detailed response below.
>
> ---
>
> > Distribution of quantile labels — class imbalance? Exact splits and sampling details.
>
> ***Re:*** We thank the reviewer for this concern. Since our goal is to evaluate the long-range dependency in the dataset, we need to exclude confounding factors (e.g., class imbalance) that influence the models’ performance.
>
> To achieve this, we first calculate the 16-hop eccentricity for all nodes on our four city networks, where their distributions are presented in Figure 5 (Appendix B.2). We then split the signals into 10 equal quantiles as node labels for classification, which ensures a balance across classes.
>
> In our experiment, we follow the common strategy in transductive node classification and randomly split the node indices by 10%/10%/80% as the train/validation/test sets. We provide masks for each for reproducibility.
>
> We will make sure this information is clear in the revised version.
>
> ---
>
> > The dataset is transductive; how will the ideas generalize to inductive settings?
>
> ***Re:*** The proposed influence measure is applicable in both transductive and inductive settings, despite our City-Networks dataset being transductive. In particular, we use our influence measure for PascalVOC-SP from LRGB in Table 2 and Figure 4 (Inductive setting with 11k small graphs of ~500 nodes). As explained in Section 4 (line 348), after training, we randomly sample 100 graphs from the testing set (more than 400k nodes in total) and report their average influence scores.
>
> At the dataset level, we agree that large-scale graph inductive learning is a promising future research direction. At the moment, even transductive learning with large graphs is arguably underexplored in the literature, and hence, we focus on this setup for this paper. We find that current GNNs and Graph Transformers are very computationally expensive even when running on a single large graph; therefore, we expect it to be even more challenging to train them on large-scale inductive learning graph tasks (in the sense that both the graph size and the number of graphs are large, and to our knowledge no such dataset exists in benchmarking of GNNs so far). This will likely require the development of more efficient models in the future.
>
> Nevertheless, a possible way to construct a large-scale inductive dataset is to sample many cities’ road networks via OpenStreetMap (or similar geographic graph sources) and define a graph-level classification or regression task on them, e.g., urban (graph) morphology classification.
>
> We will clarify this information with more details in the revised version.
>
> ---
>
> > Although the authors argue against pure geographical dependence, node features include latitude & longitude and spatially-derived attributes. It is not fully demonstrated that models can't rely largely on spatial features alone. Stronger evidence is needed to show that spatial coordinates alone cannot solve most of the signal.
>
> ***Re:*** We thank the reviewer for noticing the roles graph structure and spatial features played in our task. In fact, this is one of the key difficulties we met during our dataset construction: making sure the task requires both structural and spatial information, and fails if only one is presented. We explained and justified our detailed methods to achieve this objective in Section 2.2.
>
> As an ablation study, we use an MLP on all four city networks in our experiments in Section 3, which only uses node features such as geographical coordinates, land use, etc. Note that while edge features are augmented to node features by one-hop aggregation during dataset construction, the MLP is not capable of handling graph structural information itself. The result in Figure 3 shows a significant performance gap between MLP and other graph models, indicating that using spatial information alone is insufficient for our task.
>
> For additional experimental results, please see our replies to the next question below.

---

> ### Author Response · Authors · 2025-11-20
> **Author's Rebuttal to Reviewer ckYP (2/3)**
>
> > How sensitive is performance to the inclusion of geographic features (lat/long)? Can you report results where positional coordinates are removed? Can models exploit spatial coordinates alone (that is, consider separately MLP with coordinates only and GNNs with coordinates masked)?
>
> ***Re:***  To show the sensitivity of the baseline models to geographic coordinates, we test two GNNs (GCN and GraphSAGE) and two GTs (Exphormer and SGFormer) on Paris and Shanghai, with coordinates masked in node features (under the same experimental settings in our paper). Compared to the original results, we can observe a slight performance drop across both GNNs and GTs. At the same time, we further test MLP on these two cities with coordinates only, where the results indicate that geographical coordinates alone are not sufficient for modelling our long-range signal.
>
> | Paris | All Features | Coordinates Masked |
> | ------------- | ------------ | ------------------ |
> | **GCN**       | 53.2 ± 0.3   | 51.4 ± 0.4         |
> | **GraphSAGE** | 54.6 ± 0.2   | 52.3 ± 0.3         |
> | **Exphormer** | 55.1 ± 0.8   | 53.5 ± 0.4         |
> | **SGFormer**  | 52.0 ± 0.8   | 51.3 ± 0.7         |
>
>
> | Shanghai | All Features | Coordinates Masked |
> | ------------- | ------------ | ------------------ |
> | **GCN**       | 62.1 ± 0.2   | 61.3 ± 0.4         |
> | **GraphSAGE** | 68.3 ± 0.5   | 66.5 ± 0.4         |
> | **Exphormer** | 70.2 ± 0.4   | 67.4 ± 0.5         |
> | **SGFormer**  | 64.1 ± 0.3   | 62.8 ± 0.4         |
>
>
> | MLP | All Features | Coordinates Only |
> | ------------ | ------------ | ---------------- |
> | **Paris**    | 25.5 ± 0.4   | 12.5 ± 0.5       |
> | **Shanghai** | 28.4 ± 0.6   | 15.2 ± 0.7       |
>
> ---
>
> > What is the runtime overhead of Jacobian measurement on a single city graph?
>
> ***Re:*** As explained in Appendix D, the computation of our measurement requires $O(N \bar{N}_H)$ gradient calculations, where $\bar{N}_H$ is the average size of the H-hop ego-network in the underlying graph. As explained in Appendix D, given the large scale of our city networks, we employ a stochastic approximation that samples $N=10k$ nodes, where the computation finishes under 30 minutes for all baselines with an NVIDIA RTX 3090 GPU and 48 AMD Ryzen 3960X CPU cores. For reference, we present the runtime (in min) of four baselines in the table below.
>
> | Runtime (min) | Paris | Shanghai | La | London |
> | ------------- | ----- | -------- | -- | ------ |
> | **GCN**       | 15    | 15       | 16 | 16     |
> | **GraphSAGE** | 15    | 16       | 17 | 17     |
> | **Exphormer** | 24    | 23       | 25 | 25     |
> | **SGFormer**  | 27    | 28       | 28 | 28     |
>
> Note that while the number of gradient computations is identical across models, the total runtime also depends on the model’s architecture, since a more complex model requires more computation in its Jacobian calculation. Therefore, we can observe that the runtime for GTs is generally longer than that of the GNNs.
>
> ---
>
> > Can you please provide a Table with accuracy results you presented in Figure 3 for k=16? It will be more easier to compare the results.
>
> ***Re:*** We are glad that the reviewer mentioned this question. As stated in Section 3 (line 287), we summarize the best baseline performance (all achieved at k=16) on City-Networks in Appendix C.4 (Table 5) due to the page limit in the main text. We will make this information clearer in the revised version.

---

> ### Author Response · Authors · 2025-11-20
> **Author's Rebuttal to Reviewer ckYP (3/3)**
>
> > How would your model perform on heterophilic datasets (Texas, Wisconsin, Cornell, Roman-Empire, etc)
>
> ***Re:*** We thank the reviewer for the question regarding heterophilic datasets.
>
> In the introduction (line 57), we state that the classification tasks on heterophilic graphs are not necessarily associated with long-range dependency. As an empirical validation, we have tested models (Section 3) and computed the influence measure (Section 4) on one representative large-scale heterophilic dataset, Amazon-Ratings (24K nodes), which has a homophily score of 0.38. Our results show that all the models behave relatively short-ranged on Amazon-Ratings compared to City-Networks, as illustrated in Table 2 and Figure 4.
>
> To further support this, we present the following additional results on Texas, Wisconsin, Cornell, and Roman-Empire (as mentioned by the reviewer) in the table below. Note that Texas, Wisconsin, and Cornell are small heterophilic graphs of ~ 200 nodes and all have a small diameter of 8. We can observe that the average sizes of the influence-weighted receptive field are much smaller in the heterophilic datasets compared to those on our city networks across different models, which is consistent with the findings in our paper.
>
> | Model        | Paris | Shanghai | LA   | London | Amazon-Ratings | Roman-Empire | Wisconsin | Texas | Cornell |
> | ------------ | ----- | -------- | ---- | ------ | -------------- | ------------ | --------- | ----- | ------- |
> | **gcn**      | 4.86  | 5.55     | 5.36 | 6.09   | 1.92           | 2.04         | 1.28      | 1.11  | 1.54    |
> | **sage**     | 4.92  | 5.73     | 5.44 | 5.97   | 1.80           | 1.89         | 0.97      | 0.89  | 0.96    |
> | **gps**      | 8.18  | 7.88     | 7.92 | —      | 6.86           | 6.37         | 1.03      | 0.77  | 0.98    |
> | **sgformer** | 3.75  | 4.25     | 4.03 | 4.01   | 2.46           | 2.33         | 0.14      | 0.24  | 0.17    |
>
> ---
> \
> We’d like to thank the reviewer again for the insightful comments, and we will make sure to include the above discussion in the revised version. We believe that we have now addressed all the concerns from the reviews, and hope the reviewer could kindly consider increasing the rating in light of this.

---

> ### Author Response · Authors · 2025-11-28
> **A Kind Reminder of Our Rebuttal**
>
> Dear Reviewer ckYP,
>
> As the rebuttal period deadline approaches, we would greatly appreciate it if you could provide any further feedback on our rebuttal above so that any remaining concerns can be discussed and addressed. We sincerely appreciate your time and effort.
>
> Best regards,
>
> Authors of Submission 6143

---

### Author Response · Authors · 2025-12-01
**Rebuttal Summary**

We want to first thank all the reviewers for their detailed and valuable feedback, which we find very helpful in improving our work. We are also glad to see that they acknowledged the importance of our new long-range dataset (all reviewers), the significance in the experiment results (ckYP, e2fb), the novelty of our influence measurement (NW7j), the clarity in our presentation (ckYP, NW7j), and the soundness of our theoretical analysis (all reviewers). Here, we summarize the questions from the reviewers and our responses during the rebuttal period below.

---

### ***Reviewer ckYP***
The following questions raised by reviewer ckYP were partially addressed in the original submission, and we have clarified them further in the revised version.

- Distribution of quantile labels, class imbalance issue, and dataset split information.

  *Re:* The information is presented in Section 3 and Appendix B.2.

- Model's dependency on spatial features.

  *Re:* We tested an MLP with node features only on all four city networks in Section 3.

- A Table with the accuracy results in Figure 3 at k=16.

  *Re:* We provide the table in Appendix C.4.

- Results on heterophilic datasets.

  *Re:* We tested baselines on Amazon-Ratings in our paper, and we have added results for more heterophilic datasets as requested by the reviewer.

In addition, we have included new experiments and discussions addressing the following points in the revised version:

- The runtime of the proposed measurement on our city networks. (Appendix D)
- Discussion of possible extension to inductive settings. (Appendix E)
- Sensitivity of baselines' performance to geographic features. (Appendix B.3)

---

### ***Reviewer e2fb***
The reviewer's main concerns from the initial review and the follow-up comment are summarized as follows, which have been detailly discussed in our replies and properly addressed in the revised manuscript.

- Interpretation and the importance of our measurement.

  *Re:* We have modified the description of our measurement in the introduction, and clarified its irreplaceable role in quantifying and comparing long-range dependency across different tasks in our rebuttal.

- Insights from the benchmark and realism of the labeling approach.

  *Re:* We provide more details in the introduction about the challenges brought by our dataset, and discuss our consideration on the balance between being fully realistic and being controllable over the task range in Appendix B.2.

- Deeper models and limitations of the proposed measurement.

  *Re:* We provide additional results for deeper models at depth [32, 48, 64] in Appendix C.3, and discuss the potential limitation of our measurement caused by the bias in the underlying model in Appendix E.

- A Noise-free synthetic setting where the signal is positioned at a specific hop.

  *Re:* We added the RingTransfer experiment in Appendix C.5 and discussed the behavior of our measurement with different models, and the reviewer agrees with our analysis.

- How our theoretical analysis of over-smoothing relates to the Braess paradox and previous work.

  *Re:* We provide a detailed discussion in Appendix F.4, in which the reviewer agrees with our conclusions.

Please see our replies below for more details.

---

### ***Reviewer NW7j***

We have clarified the following information as suggested by the reviewer in the revised version.

- A more detailed comparison to a previous metric with reference in the introduction. (Section 1)
- A more concrete statement in the main text regarding the exact list of features. (Section 2)
- A more precise description of the arguments made by the authors of the LRGB paper. (Section 1)

---

We believe all the questions and concerns from the reviewers have now been fully addressed, and we hope this will affirm the reviewers' positive evaluation of our work.

---

### Meta-Review · Area_Chair_nCtn · 2026-01-07

**Summary:**

This work introduces a dataset based on city road networks of several cities. Given this graph, the target task is to compute local node eccentricities, which requires long-range information. Also, the authors propose an influence measure for the quantification of long-range dependencies (for a given model).

The reviews are overall positive. Based on the reviews and discussion, I am leaning towards acceptance of the paper.

My main concern, however, is that the proposed dataset is semi-synthetic. The graph structure and node features are realistic, but the task is synthetic and graph-based (the target is computed using the graph structure and road lengths). To make the target nontrivially depend on node features, the authors convert edge features into node features via neighborhood aggregation, which makes road length not directly available. With such aggregation, a lot of information is lost, which makes the dataset less realistic (in practice, edge features are available and can be directly used). The graph is also made undirected, which additionally reduces the amount of available information. Also, the considered problem is originally a regression problem, but it is converted to classification via discretization. Thus, I would say that the dataset is suitable for the purposes of this paper (analysis of long-range dependences), but it is semi-synthetic and has a limited potential to be used as a realistic benchmark beyond this research topic. When considering the dataset contribution, this dataset can be positioned (in my opinion) between completely synthetic datasets designed specifically for the analysis of long-range tasks (like GLoRa) and road network datasets that are more realistic but do not aim at evaluating long-range properties (like Velikonivtsev et al., arXiv:2510.02278).

**Reviewer Concerns:**

Some of the concerns raised by the reviewers are:
- Long-range influence metric is not model-agnostic. To address this, the authors changed the wording and clarified this in the paper.
- Several limitations of the measure have been discussed. The authors added the corresponding comments to the paper and also conducted additional experiments.
- The defined task is synthetic and is not realistic. The authors conducted an additional experiment during the discussion period and also explained that they aim to balance between realistic and controllable settings. But in general, the concern remains.
- Reviewers asked for a comparison on synthetic benchmarks. The authors added the RingTransfer experiment.
- Reviewers asked for  a more detailed comparison with Bamberger et al. (2025). The authors updated the paper accordingly.

**Reviewer Scores:**

The initial scores are 8 (NW7j), 6 (e2fb), 4 (ckYP). I think that 4 could increase to 6 since the concerns have been addressed in the rebuttal. Some of the concerns of Reviewer e2fb have also been addressed, but I cannot estimate whether the score would increase or stay the same.

---

### Decision · Program_Chairs · 2026-01-26

Accept (Poster)